



# Trace metal distributions in the transition zone from the Greenland Ice-Sheet to the surface water in Kangerlussuaq fjord (67 °N)

Clara R. Vives[1], Jørgen Bendtsen[1], Rasmus D. Dahms[1], Niels Daugbjerg[2], Kristina V. Larsen[1], and Minik T. Rosing[1]

[1]Centre for Glacial Rock Flour Research, Globe Institute, University of Copenhagen, Copenhagen K, Denmark
[2]Marine Biological Section, Department of Biology, University of Copenhagen, Copenhagen Ø, Denmark

**Correspondence:** Clara R. Vives (clara.vives@sund.ku.dk)

**Abstract.**

Glacial rock flour (GRF), an ultra-fine sediment formed beneath glaciers, contains high concentrations of silicate and trace metals, including iron (Fe) and manganese (Mn). In Greenland, meltwater discharge transports approximately 1.28 Gt of suspended sediments annually into the oceans, significantly influencing trace metal concentrations and marine biogeochemical cycles. This study investigates the spatial distribution of trace metals, nutrients and suspended sediment concentrations (SSC) from the Russell Glacier at the Greenland Ice Sheet, through the Akuliarusiarsuup Kuua meltwater, and into the Kangerlussuaq fjord in western Greenland. Dissolved trace metals were relatively high in the river and low-salinity surface waters in the fjord, showing that the fjord acts as an important source of trace metals to the marine environment. However, trace metal concentrations, particularly Fe and zinc (Zn), exhibited significant non-linear decreases beyond salinity levels of 14, underscoring the complex processes affecting trace metal supply from rivers to fjords and coastal waters. In contrast, silicate concentrations increased in river water due to weathering of GRF and decreased gradually in the inner-fjord due to mixing with surface water. Uranium (U) and molybdenum (Mo) were undetectable along the river but increased in the fjord, indicating that these elements primarily originate from the ocean. These findings highlight the complex interplay of physical, chemical, and biological processes regulating trace metal and nutrient dynamics in glacier-influenced fjord systems, with implications for primary productivity and carbon cycling in polar oceans.

## 1 Introduction

Nearly all glaciers in Greenland have been thinning or retreating in the past few decades, and the Greenland ice sheet (GrIS) has experienced an acceleration in calving events since 1985 (Chen et al., 2006; Zeising et al., 2023; Greene et al., 2024). Due to this mass loss driven by increased surface melting and enhanced discharge from outlet glaciers (IMBIE, 2020), the GrIS is one of the largest contributors to contemporary sea level rise globally (Bamber et al., 2019), adding an average of 0.77 mm yr$^{-1}$ (Clark et al., 2015), that is 21% of the global mean since 1993 (WCRP, 2018). Half of the ice mass loss in polar zones over the last decades resulting from warming is caused by ice-sheet discharge through the marine-terminating glaciers (Shepherd and Wingham, 2007; Slater et al., 2021). Climate models suggest that continued ice loss in Greenland will accelerate (Khan et al.,





2022), leading to an increased contribution of global mean sea level rise from the GrIS by 2100 (Goelzer et al., 2020; Edwards
et al., 2021; Aschwanden and Brinkerhoff, 2022; Paxman et al., 2024).

The debris from glacial erosion, which contributes to approximately 7-9% of the global sediment flux to the sea (Overeem
et al., 2017), contains fine sediments which remain in suspension and are transported into the coastal ocean waters. In Green-
land particularly, meltwater transports ∼0.6 - 1.3 Gt of suspended sediments into the oceans every year (Hasholt et al., 2006;
Overeem et al., 2017; Hawkings et al., 2017) and where fine particles dominate the sediment. These ultra-fine sediments origi-
nating beneath the GrIS from the erosion of the bedrock by glacial movement are referred to as glacial rock flour (GRF). This
heavy physical erosion makes the GRF chemically immature compared to more weathered sediments, and its mineralogical
composition is therefore very similar to its source rocks. The composition of GRF varies depending on the bedrock source, but
in Southwest Greenland a common suite of present minerals is quartz, feldspars, phyllosilicates, and amphiboles. GRF is also
a felsic silicate mineral which contains a suite of trace metals, e.g., including iron. Mercury concentrations have been found to
be very low in meltwater entering Kangerlussuaq fjord (Jørgensen et al., 2024) in accordance with the analyzed composition
of GRF (Sarkar, 2021). Sarkar (2021) also found that other toxic substances in GRF from different locations around Greenland
were present in very low concentrations.

While it is established that glacial discharge and sediments from glacial erosion is primarily transported through rivers
and into the fjords, the mechanisms governing the transition of trace metal inputs, particularly from glaciers and glacial rock
flour, into marine systems remain unclear. Trace metals, such as iron, are critical micronutrients for phytoplankton growth
and biological processes, thereby playing an essential role in the global carbon cycle. For example, iron is a vital component
in photosynthetic proteins involved in the electron transport chain and redox reactions within the Photosystem II apparatus
(Strzepek et al., 2012; Raven et al., 1999). In High Nitrate Low Chlorophyll (HNLC) regions like the Southern Ocean and parts
of the North Pacific and North Atlantic, trace metal availability is a limiting factor for primary productivity and subsequent
carbon export (Boyd and Ellwood, 2010). Hence, alongside macronutrients such as phosphate, nitrate, and silicate, trace metals
regulate oceanic biological production.

Other iron sources such as aeolian iron or dust support primary production in regions of the Northeast Atlantic (Blain et al.,
2004). However, previous studies have found that aeolian sources are not sufficient to support primary production in the North
Atlantic during spring and summer (Moore et al., 2006). Other studies have also found that regions of the North Atlantic
become iron limited in the summer (Nielsdóttir et al., 2009; Ryan-Keogh et al., 2013; Browning and Moore, 2023). Greenland
glacial meltwater has been identified as a significant source of bioavailable iron (Hawkings et al., 2014) and silica (Hawkings
et al., 2017), with an estimated annual flux of dissolved and potentially bioavailable particulate iron to the North Atlantic Ocean
of approximately 0.3 Tg (Bhatia et al., 2013). Understanding the fluxes and transformations of these trace metals from glacial
systems to oceanic environments is crucial for predicting their contribution to marine biogeochemical cycles under ongoing
climate change. However, field data and a more complete understanding of processes acting on the dissolved suspended material
from the glaciers through transitioning rivers and into the fjords are lacking, particularly for a wider range of trace metals and
macronutrients like nitrate, silicate and phosphate.





Furthermore, GRF is considered as a potential means of action for marine Carbon Dioxide Removal (mCDR). Weathering of GRF supply the ocean with macronutrients and trace metals, supporting phytoplankton growth (Bendtsen et al., 2024).

Additionally, GRF has been shown to increase the effects of enhanced rock weathering on land, due to its high silicate content (Gunnarsen et al., 2023; Dietzen and Rosing, 2023). Its dissolution in seawater supplies essential micronutrients, e.g., trace metals, like iron and manganese, which can alleviate nutrient limitations and promote carbon sequestration. However, the impact from many different processes, e.g., adsorption, flocculation and scavenging, affects the distribution of GRF. Therefore, this study also describes the distribution of GRF in its natural environment motivated by its potential future usage as a source

for mCDR.

In this study we investigate the distributions of trace metals and macro-nutrients in the transition zone from the Russell Glacier at the Greenland Ice Sheet, through the Akuliarusiarsuup Kuua meltwater, and into the inner part of Kangerlussuaq Fjord. Meltwater and seawater samples were analysed for trace metals, nutrients and other environmental variables, e.g., salinity and suspended sediment. Finally the role of internal sinks, e.g., adsorption, and biological uptake, for the transport of GRF-

related trace metals to the open ocean is discussed.

## 2 METHODS

### 2.1 Study area

Samples were collected during the Glacial Rock Flour in the Sea (GROFS1) field campaign, from 20 June to 6 July 2023. Sampling was carried out in the inner part of the Kangerlussuaq fjord located in west Greenland ($\sim$67 °N) and along the

meltwater river (Akuliarusiarsuup Kuua, eng. Watson river) that enters the bottom of the fjord (Fig. 1). The river originates at the Russell glacier; an ice tongue from the GrIS. Water was sampled at 35 stations that covered the transition zone from the Russell glacier, along the Akuliarusiarsuup Kuua and into and along the inner part of Kangerlussuaq fjord. The river receives runoff from three major glacier tongues along its passage from the Russell glacier and towards the fjord (Hasholt et al., 2018). The town and airport of Kangerlussuaq are located at the outlet of the river that enters the fjord via a $\sim$10 km long river

delta. The inner fjord covered the area from the river delta and about half-way into the inner basin of Kangerlussuaq fjord. Kangerlussuaq fjord is a 100 km long fjord and is separated into a deep inner and relatively shallow outer bassin. The two basins are separated by a narrow and shallow strait. The inner basin is about 80 km long and in general deeper than 200 m and with a maximum depth of $\sim$275 m (Nielsen et al., 2010).

The inner basin receives freshwater from the Akuliarusiarsuup Kuua and Umivit river. The Akuliarusiarsuup Kuua is passing

below the Kangerlussuaq bridge (eng. Jack T. Perry Memorial bridge) at Kangerlussuaq city and the annual discharge has been observed to vary between 3.8-11.2 km$^3$ in the period 2006-2017 (van As et al., 2018). Runoff starts in spring and obtain peak values of $\sim$2000 m$^3$ s$^{-1}$ in July. In 2023 the sea ice was reported to break up in early June and the spring was relatively dry compared with the previous 10 years (Fig. 2). Hence, runoff in early June were relatively modest and the estimated accumulated runoff from the river by the end of June was 0.2 km$^3$ (van As, 2022). However, a relatively large river transport was visually

observed at the outlet at Kangerlussuaq town during the entire field campaign. The discharge from the Umivit river has been





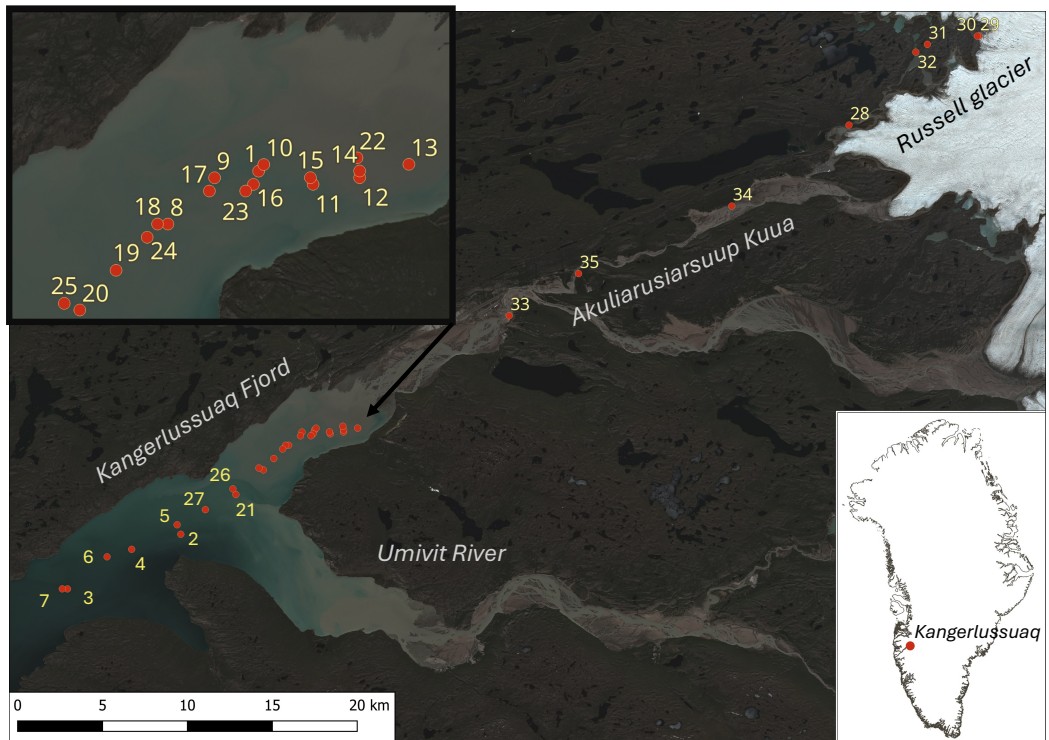

**Figure 1.** Study area in Kangerlussuaq fjord and stations for water sampling in the fjord and in the river towards the GrIS. Station numbers are shown in the upper left inlet and the location of the fjord system is indicated on the map of Greenland. The Sentinel-2 satellite image is from 2 July 2023.

estimated to be ∼70 % larger than the discharge from the Akuliarusiarsuup Kuua (Monteban et al., 2020). The area off the fjord arm with the Umivit river outlet that connects to the main fjord was located at a distance of ∼ 50 km from the glacier, i.e. ∼10 km from the Akuliarusiarsuup Kuua outlet (Fig. 1).

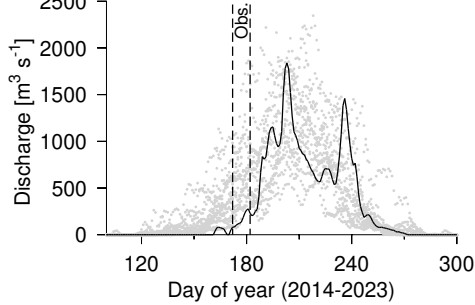

**Figure 2.** Discharge from the Akuliarusiarsuup Kuua (van As, 2022) in the period 2014-2022 (light gray bullets) and in 2023 (black line). The period of the field campaign is indicated (dashed lines).





Samples were collected from 27 stations in the fjord, 6 stations along the river, and 2 stations in proximity (0-200 m) of the
glacier (Table A1). The total transect from the glacier to the outermost station in the fjord was ∼65 km. A detailed synoptic
transect was made from a station in the innermost part of the fjord and within a distance of ∼5 km from the river delta and
∼6 km into the fjord. This transect was made within 2 hours and vertical profiles were made at the end-stations (st. 13 and 20)
and surface samples (0-1 m depth) were collected at stations in between. The transect covered an area with a visible change
of suspended material from the river plume and the synoptic transect ended in the middle of the fjord and off the Umivit fjord
arm.

## 2.2   Temperature, salinity and freshwater content

Measurements of conductivity (C), temperature (T) and pressure, i.e. ∼depth (D), were made with a loose-tethered free-
fall Rockland Scientific International (RSI) VMP-250 microstructure vertical profiler in the upper ∼150 m. The profiler was
equipped with a conductivity, temperature and pressure sensor (JFE Advantech) that operated at 16 Hz. In addition, an under-
way CTD probe (Ocean Science, Seabird CTD) was applied at stations where water samples only were made in the upper 10
m (st. 23 - 26). Measurements from the VMP-CTD and the UCTD were in accordance and a comparison at 40 m depth showed
a difference less than 0.1 °C and 0.1 g kg$^{-1}$ for temperature and salinity, respectively.

Temperature, salinity and density are reported as conservative temperature ($\Theta$), absolute salinity ($S_A$) and potential density
anomaly ($\sigma_\Theta$), respectively (IOC et al., 2010). The correction factor for absolute salinity is not well described in coastal waters
and fjords around Greenland (Bendtsen et al., 2021) and was therefore set to zero (i.e., $\delta S_A = 0$). Temperature and salinity
samples were binned in 0.5 m intervals from a depth of 0.5 m.

In addition to surface salinity the freshwater content ($F_W$) of the upper part of the water column can be applied for analysing
the impact from runoff. The freshwater fraction represents the amount of freshwater needed to dilute a water column with a
certain reference salinity and of a given depth to obtain the observed salinity. Thus, the freshwater fraction is an integrated
measure of the amount of freshwater in the upper layer. Hence, it represents an integrated impact from freshwater sources
(e.g., runoff, precipitation and sea ice melt) over a time period and is therefore not so sensitive to temporal or spatial variation
of the surface salinity (Bendtsen et al., 2014). The freshwater content was related to a reference salinity ($S_0$) of 24.2 g kg$^{-1}$
representative for the salinity at the bottom of the surface layer (D = 15 m) and calculated as:

$$F_W = \int\limits_{-D}^{0} \frac{S_0 - S}{S_0} dz$$


The integral is made along the vertical axis (z) and $F_W$ was calculated for each station.

The near surface salinity (0.5 - 1 m) at stations along the synoptic transect between station 13-20 were interpolated between
salinities at the end stations, i.e. st. 13 ($S_A = 6.51$ g kg$^{-1}$) and st. 20 (6.62 g kg$^{-1}$). Thus, surface salinity showed a minor change
along the transect (Fig. 3c). Similarly, the freshwater content was interpolated along the transect between the corresponding





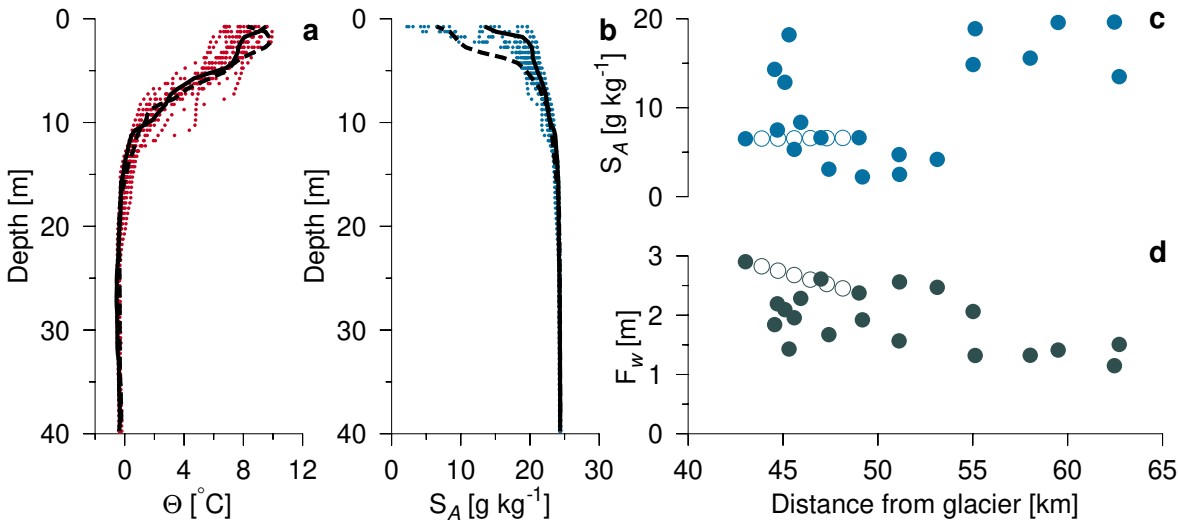

**Figure 3.** a) Conservative temperature (red, $\Theta$) and, b) absolute salinity (blue, $S_A$) measured in the inner part of Kangerlussuaq (st. 1-27). Stations near the Akuliarusiarsuup Kuua outlet and the furthest distance from the river are shown with dashed and full lines, respectively. c) Surface salinity (0.5 - 1 m) and d) freshwater content ($F_w$) from stations in the fjord. Interpolated salinity and freshwater content at the synoptic transect between station 13-20 are marked by hollow circles.

values of $F_W$ = 2.90 m (st . 13) and 2.38 m (st. 20). The change in Fw of ∼0.5 m reflected that the depth of the surface plume decreased along the transect into the fjord (Fig. 3b, Table A1).

### 2.3 Dissolved trace metals

Trace metal samples were collected at 28 stations at the surface, with trace-metal clean bottle equipment. Samples were collected from undisturbed water in the fjord and in the river. One station (st. 33) was located beside a bridge at the entrance
to the river delta and before the discharge from the small town of Kangerlussuaq. Samples in the river were collected facing the current to avoid contamination. All equipment used to sample the concentration of trace metals in the water was prepared following the GEOTRACES protocol (Cutter et al., 2010). Low-density polythelene (LDPE) bottles were washed in Decon for 2 weeks before being placed in an acid bath (HCl 6M) for an additional 4 weeks. All bottles were then rinsed 3 times with ultra-pure MilliQ water and triple bagged for transportation. To measure dissolved concentrations, aliquots were syringe-
filtered through acid-washed Pall Acropak Supor capsule filters (0.2 mm), then acidified with ultrapure 65% $HNO_3$ to a final concentration of 2%. Dissolved trace metals were then analysed on an ICP-OES at the Sustain Lab (Danish Technical University, Denmark). Each sample was analysed 3 times. The detection limit was calculated for each element and each run from the calibration curve in the instrument, using a pre-made solution and certified standards. Background levels for each element and each sample were subtracted from the final values. Quantification limits for each element are listed in Table 1.





**Table 1.** Mean and standard error for concentrations ($\mu$g L$^{-1}$) of dissolved trace metals. The transect is divided into five distinct areas: glacier, river, river delta, inner fjord (low salinity < 13) and fjord (high salinity). Dissolved trace metals included are: iron (Fe), manganese (Mn), cobalt (Co), copper (Cu), zinc (Zn), nickel (Ni), molybdenum (Mo), arsenic (As), vanadium (V), and uranium (U). Values below quantification limit (QL) are shown as the QL value ($\mu$g L$^{-1}$) for each element respectively, and marked with an asterix (*). A table including all trace metal samples is included in Table A2.

| | Glacier ($n = 1$) | River ($n = 5$) | River delta ($n = 1$) | Inner fjord ($n = 19$) | Fjord ($n = 6$) | Quantification limit |
|---|---|---|---|---|---|---|
| **dFe** | 17.76 | 47.16±36.92 | 36.67 | 19.61±27.72 | 3.75±1.21 | 5 |
| **dMn** | 9.4 | 7.21±4.17 | 9.8 | 7.60±2.72 | 4.4±1.68 | 0.5 |
| **dCo** | 0.1* | 0.13±0.03 | 0.1* | 0.13±0.04 | 0.1* ±0 | 0.1 |
| **dCu** | 1.18 | 1.56±1.03 | 1.4 | 1.36±1 | 1±0.14 | 0.5 |
| **dZn** | 22.64 | 23.26±17.97 | 19.96 | 30.87± 32.35 | 7.13±5.1 | 0.5 |
| **dNi** | 0.65 | 0.61±0.34 | 0.1* | 0.79±0.16 | 0.73±0.05 | 0.1 |
| **dMo** | 0.1* | 0.14±0.05 | 0.27 | 1.03±0.77 | 3.07±1.18 | 0.1 |
| **dAs** | 0.5* | 0.5* | 0.5* | 0.5* | 0.54±0.07 | 0.5 |
| **dV** | 0.5* | 0.61±0.15 | 0.997 | 0.61±0.11 | 0.5* | 0.5 |
| **dU** | 0.1* | 0.1* | 0.1* | 0.21±0.16 | 0.60±0.29 | 0.1 |

**2.4   Nutrients and suspended sediment**

Water samples for nutrients, and suspended sediment concentration (SSC) were collected using a 5 L Niskin bottle operated by a messenger. Water samples were collected at standard depths in the upper 40 m. Samples for nutrients were filtered through a syringe filter (Filtropur S, polyethersulfone PES-membrane, pore size 0.2 $\mu$m) and immediately placed in a cooler box (<0°C)





and stored at -20 °C on land within 6 hours. The samples were analyzed for nitrite, nitrate, ammonia, phosphate and silicate by
wet-chemistry methods (Grasshoff, 1983) with detection limits of 0.04, 0.1, 0.3, 0.06 and 0.2 $\mu$M, respectively (DCE, Aarhus
University, Denmark). Suspended sediment concentrations (SSC) were determined from 1-5 L water samples filtered through
a 200 $\mu$m mesh on site and 0.3 $\mu$m glass fiber filters (Advantec GF-75 ø 47 mm) and subsequently dried for $\approx$ 12 hours at 60
°C. Samples were transported in Ziploc bags and weighted on a Mettler Toledo MS205DU Analytical balance.

## 3  RESULTS

### 3.1  Stratification, salinity and freshwater content

The vertical stratification in the fjord was characterized by a shallow thermo- and halocline in the upper 3-5 m (Fig. 3). At the
innermost station (st. 13) the salinity increased from 6.48 g kg$^{-1}$ at 1 m depth to 10.73 g kg$^{-1}$ at 3 m depth, corresponding
to a change in density ($\sigma_\Theta$) from 4.95 to 8.12 kg m$^{-3}$. This relatively strong stratification was also present at the outermost
station (st. 7) where the corresponding vertical change in salinity (density) increased from 13.48 g kg$^{-1}$ (10.25 kg m$^{-3}$) at 0.5
m depth to 20.24 g kg$^{-1}$ (15.69 kg m$^{-3}$) at 3 m depth. Thus a strong density stratification characterized the upper few meters
in the study area. A secondary halocline down to 10-15 m depth indicated a deeper mixing of surface water had occurred after
the sea ice breakup in early June. Below 15 m depth the salinity and temperature variation was relatively small and the deeper
temperatures and salinities were relatively homogenous along the fjord transect, e.g., temperature and salinity at 40 m depth
was typically -0.39 °C and 24.38 g kg$^{-1}$, respectively. The coldest temperature was found at 26 m depth of -0.54°C ($S_A$ =
24.28 g kg$^{-1}$) and this subsurface temperature minimum indicated the pervious depth of the mixed layer during winter time
and below the sea ice.

The lowest surface salinity (2.49 g kg$^{-1}$ at st. 21) was observed off the Umivit fjord arm, and influenced by the additional
outflow from the Umivit River (Fig. 3c). In general, the salinities in the inner part of the fjord varied from 3.08 g kg$^{-1}$ and up to
18.21 g kg$^{-1}$. Relatively high salinities were observed at some stations near the innermost part of the fjord in the first week of
the field campaign. This period was characterized by a relatively low runoff (Fig. 2) and this could explain these observations.
Also wind forcing along the fjord caused visible changes of the location of the river plume. The relatively large tidal range
between 2 - 4 m (Nielsen et al., 2010) also contributed to daily variations. The freshwater content in the upper 15 m of the
surface layer showed a decrease from the inner stations with a F$_W$ $\sim$3 m to $\sim$1 m at the outermost station (Fig. 3d). The spatial
and temporal variability of F$_W$ was significantly less than observed from the surface salinity.

### 3.2  Trace metal distributions

Concentrations of Co, Cu, Fe, Mn, Mo, Ni, V, U and Zn were in general above detection limits in the inner part of the fjord and
most of these tracers were also present at detectable levels in the river (Table 1). Tracer concentrations of Cd, Cr, Pb, Se, Ti and
Tl were generally observed below the detection limits (detection limits ($\mu$g L$^{-1}$) of: 0.1, 0.5, 0.1, 1, 5, and 0.1, respectively,
Table A2).





### 3.2.1 Distributions based on distance from the glacier


The spatial distributions of trace metals were first analysed in relation to their distance from the glacier (Fig. 4). Dissolved manganese (dMn) was relatively constant in the river ($\sim$10 $\mu$g L$^{-1}$), and higher than the stations closest to the glacier ($\sim$ 3 $\mu$g L$^{-1}$). In the innermost part of the fjord (i.e., S$_A$ <13 g kg$^{-1}$) dMn exhibited a range from 5 - 12 $\mu$g L$^{-1}$. At stations with higher salinities dMn decreased to about 5 $\mu$g L$^{-1}$. Dissolved zinc (dZn) and copper (dCu) exhibited a similar spatial pattern

as dMn where the concentration in the fjord were higher in the inner part of the fjord than in the river and near the glacier. For dCu, the concentrations increased in the river when compared to the relatively constant distributions of dMn and dZn.

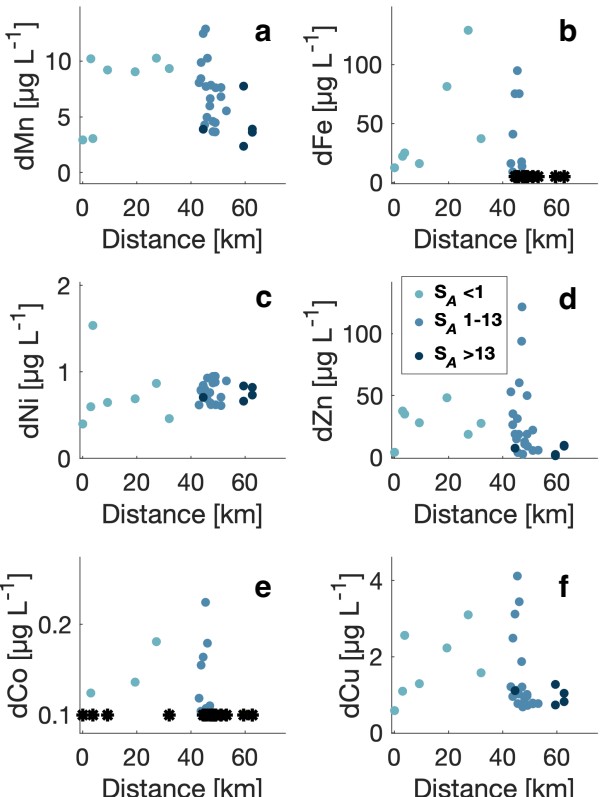

**Figure 4.** Concentration of dissolved trace metals versus distance from the glacier. Concentration below the quantification limit is shown at the detection limit for each element (*). Data points with salinities in the intervals: <1, 1-13 and >13 g kg$^{-1}$ are shown with colors.

Distributions of dissolved iron (dFe) and dissolved cobalt (dCo) exhibited a similar pattern. However, their distributions were significantly different from the other elements. In general, the dFe-concentration increased along the river from 10 to 120 $\mu$g L$^{-1}$ and from 0.12 to 0.19 $\mu$g L$^{-1}$ for dCo. In the high saline part of the transect the concentrations ranged from 5 (i.e.,



the detection limit for dFe) to 100 $\mu$g L$^{-1}$ for dFe, and 0.1 (detection limit for dCo) to 0.24 $\mu$g L$^{-1}$ for dCo. Most of the dFe-samples below the detection limits were found in the outer part of the transect (S$_A$ >13 g kg$^{-1}$).

    Dissolved nickel (dNi) showed a distinct pattern, different from the other elements, where concentrations increased slightly from the glacier to the river from 0.5 to 1 $\mu$g L$^{-1}$ (and up to 1.5 $\mu$g L$^{-1}$ for one data point). In the fjord, concentrations appeared constant with all values around 1 $\mu$g L$^{-1}$.

**3.2.2   Trace metals versus surface S$_A$**

Salinity in the inner part of the fjord reflected the location of the river plume, and distributions of the tracers versus S$_A$ therefore accounts for temporal variation of the river plume during the field campaign. Therefore the distribution were also analysed versus surface S$_A$ (Fig. 5). The distribution of trace metals against salinity exhibited significant differences between the elements. The concentrations for dMn, dCo and dNi were relatively constant compared to dFe, dZn and dCu. For dFe, the

concentrations were highest at the glacier and in the river (i.e., S$_A$ = 0) and at two stations (st. 15 and 16) in the inner part of the fjord (Table A2). Low concentrations were observed in the high-saline part of the transect. Similar patterns were observed for dZn, dCo and dCu. Mangenese showed a relatively gradual decrease from the river and into the fjord. dNi showed minor changes in the fjord.

**3.2.3   Trace metals versus freshwater content**

Similarly to surface salinity, the freshwater content reflects the position of the river. However, as F$_W$ is determined from the integrated salinity in the surface layer it is a suitable and less variable representative for the average position of the river plume in the fjord. Trace metal distributions in the fjord were therefore also analysed in relation to the freshwater content. Analysis in relation to F$_W$ was only relevant in the fjord, and therefore the river measurements were not included in relation to F$_W$.

    The highest concentrations of dMn, dFe, dZn, dCo and dCu were observed at stations with a freshwater content larger than

2.5 m (Fig. 6). That implies that these tracers had the highest concentrations at stations with the largest impact from runoff. At stations with less F$_W$ the concentration of dFe and dCo were below or close to the detection limits. The distribution of dNi was relatively constant in the fjord ($\sim$ 0.8 $\mu$g L$^{-1}$).

    Three trace metals (Mo,V and U) were analysed both in relation to distance from the glacier and F$_W$ (Fig. 7). All three tracers showed concentrations below the detection limit close to the glacier and dU was below detection limit in the entire river

(i.e., detection limits of 0.1, 0.5 and 0.1 $\mu$g L$^{-1}$ for dMo, dV and dU, respectively). Similarly, dMo was relatively low in the river whereas dV showed elevated concentrations midway between the glacier and the outlet. Concentrations of dMo and dU was low or below detection limits at stations with a F$_W$ larger than 2.5 m, i.e., stations with the largest impact from runoff. Low concentrations of dV below detection limits were observed at stations with a Fw less than 2.5 m (i.e., less impacted by runoff), however, some variability characterized its distribution along the transect in the fjord.




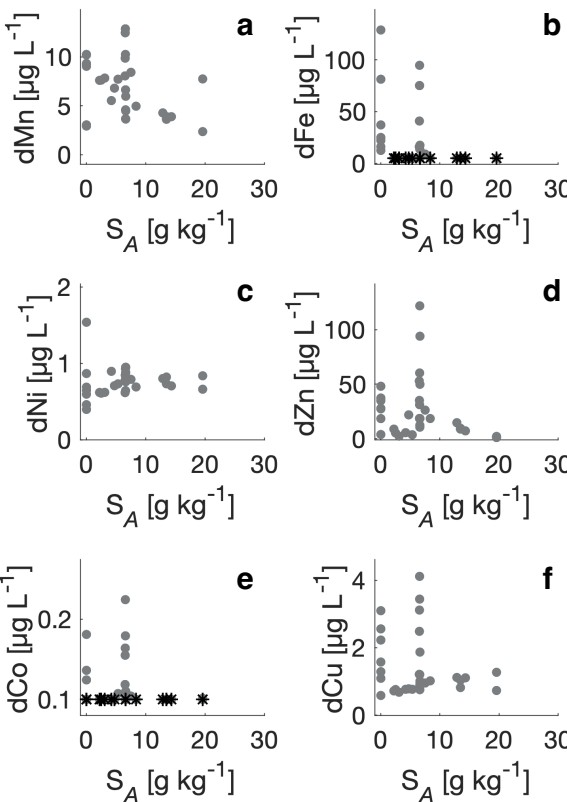

**Figure 5.** Trace metal concentrations versus surface salinity ($S_A$). Dissolved manganese (dMn), dissolved iron (dFe), dissoved nickel (dNi), dissolved zinc (dZn), dissolved cobalt (dCo), and dissolved copper (dCu) are shown consecutively in panels a-f. Measurements below detection limits are shown (*).

## 3.3 Nutrients and SSC distributions

Silicate (Si) was lowest near the glacier and the largest values were observed in the fjord and ∼40 km from the glacier (Fig. 8). Silicate concentrations increased gradually from the glacier along the river from 0 to 5 $\mu$mol L$^{-1}$, and obtained its largest value of 10 $\mu$mol L$^{-1}$ at the site before the river delta and closest to the fjord. In the inner fjord, where surface salinity was between 1-13 g kg$^{-1}$, Si ranged between 5 and 20 $\mu$g L$^{-1}$ (Fig. 8b). These values were significantly higher than near the glacier and along the river. The Si concentration decreased to 2-7 $\mu$g L$^{-1}$ at higher salinities. Phosphate and dissolved inorganic nitrogen (DIN, i.e., the sum of ammonia, nitrite and nitrate), however, showed a more variable relationship with salinity (Fig. 8d,f) with a general decrease towards higher salinities. The vertical nutrient distributions showed that silicate increased near the surface whereas phosphate and DIN showed very low concentrations in the upper 10 m of the surface layer. Profiles of nutrients showed different distributions with depth (Fig. 9): Silicate was highest at the surface and decreased below 10 m. Phosphate and DIN



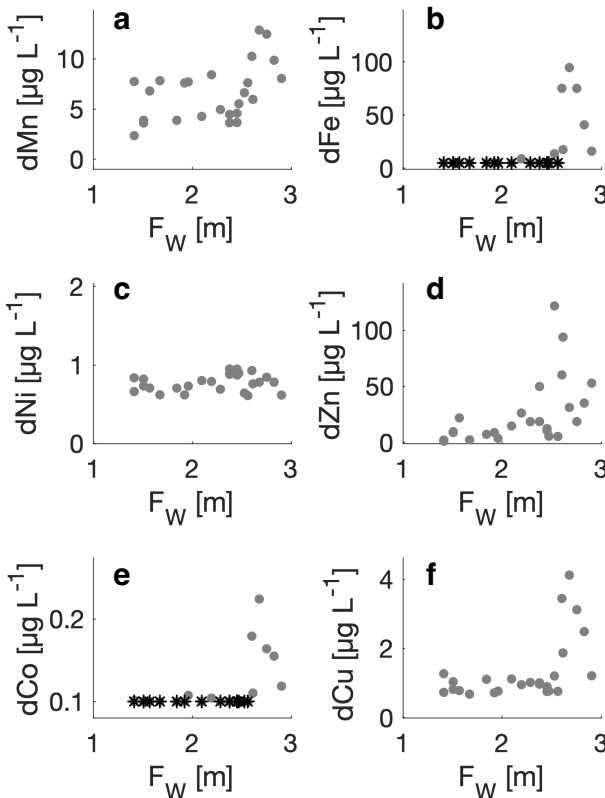

**Figure 6.** Trace metals versus freshwater content ($F_W$). Dissolved manganese (dMn), dissolved iron (dFe), dissoved nickel (dNi), dissolved zinc (dZn), dissolved cobalt (dCo), and dissolved copper (dCu) are shown consecutively in panels a-f. Measurements below detection limits are shown (*).

exhibited a similar pattern, where phosphate and DIN concentrations increased with depth and obtained deep values at 40m depth of $\sim$0.4 $\mu$mol L$^{-1}$ and 5 $\mu$mol L$^{-1}$ for phosphate and DIN, respectively.

The Suspended Sediment Concentrations (SSC) were measured at the surface and at 1 m depth (Fig. 10). SSC showed a general decrease from $\sim$100 mg L$^{-1}$ in the river and at low salinities to values of $\sim$10 mg L$^{-1}$ in the high-saline part of the transect.

## 4 Discussion

The spatial distribution of trace metals from the glacier to the fjord reflects the complex interactions and impacts from runoff, physical mixing, weathering, erosion and water-sediment fluxes, the concentration of metal-binding organic compounds, scavenging associated with flocculation, and biological uptake and remineralisation of metal-containing organic matter. Thus, the



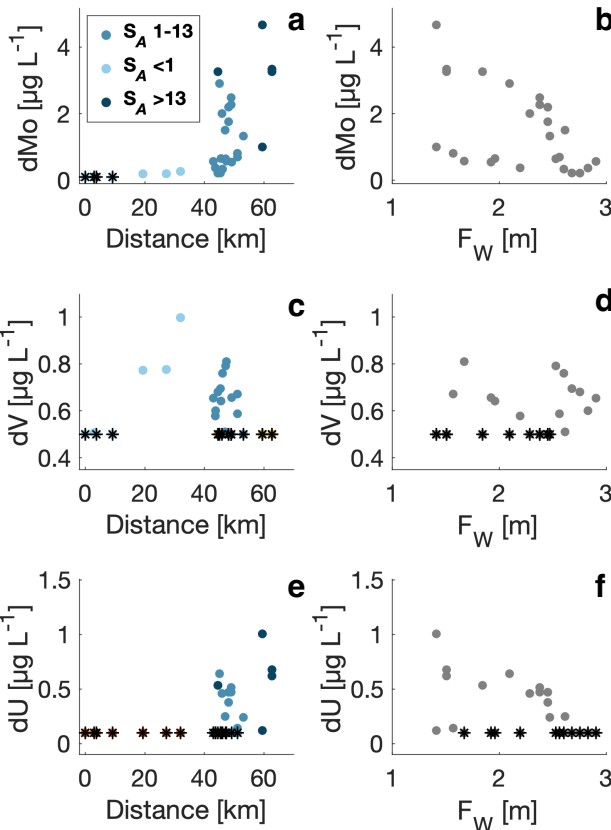

**Figure 7.** Distribution of dissolved molybdenum (dMo, a), dissolved vanadium (dV, c) and dissolved uranium (dU, e) versus distance from the glacier (km) and freshwater content in the fjord (b,d,f). Data points with salinities in the intervals: $<1$, 1-13 and $>13$ g kg$^{-1}$ are shown with colors in panels a, c, and e. Measurements below detection limits are shown (*)

flux of trace metals in the meltwater river that finally enters the fjord and coastal ocean is heavily influenced by many different

processes in the transition zone.

## 4.1   Tracer gradients from glacier to fjord

The concentration of dFe showed relatively high values close to the glacier (13 $\mu$g L$^{-1}$) and a tendency to increase along the river (Fig. 4b). The highest value (129 $\mu$g L$^{-1}$) was observed mid-way between the glacier and the river-delta and this suggest that significant dFe-sources were present along the river (Table A2). This could be due to enhanced weathering and

metal-mobilization from deposits of glacial rock flour along the river interacting with the strong meltwater current. Similar high dFe-concentrations have been observed near glaciers from the Greenland Ice sheet. Bhatia et al. (2013) reported values in the range of 21-56 $\mu$g L$^{-1}$ from glaciers located $\sim$100 km north of our study site. Zhang et al. (2015) observed correspondingly





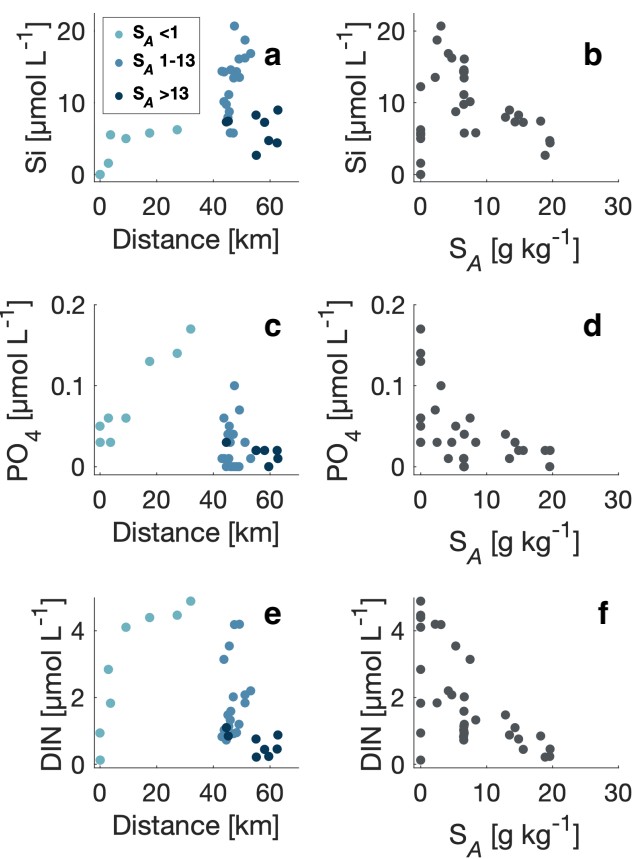

**Figure 8.** Distributions of (a) silicate, (c) phosphate, and (e) dissolved inorganic nitrogen (DIN) concnetrations ($\mu$mol L$^{-1}$) versus distance from the glacier (km). The salinity gradient follows a scale of blue colors, where lower salinity is shown in lighter blue and higher salinity in darker blue. Panels b, d, f show the spatial distributions of the same macronutrients based on the salinity gradient.

high values (6-45 $\mu$g L$^{-1}$) in the proximity of a glacier on Svalbard. Thus, the high concentration near the glacier and in the river is in general accordance with previous findings that glacier meltwater may carry high concentrations of dFe towards

the sea. Concentrations in the fjord showed a significant decrease of dFe near the river outlet and it was associated with the location of the river plume. The highest concentrations were observed in areas close to the river where the freshwater content in the upper 15 m was above 2.5 m. Further from the river outlets and where the freshwater content was below 2.5 m the concentration was below the detection limit. Previous studies have correspondingly identified the river-seawater transition as an area with a large sink of dFe (Boyle et al., 1977; Zhang et al., 2015). The distribution in the inner fjord showed an elevated

concentration off the fjord arm that receives meltwater from the Umivit river. The additional runoff from the Umivit river and the combined mixing from the two river outlets may explain the increased concentration of dFe in the inner part of the fjord where the freshwater content was above 2.5 m (Fig. 6). The distributions also indicated a relatively high dFe-concentration



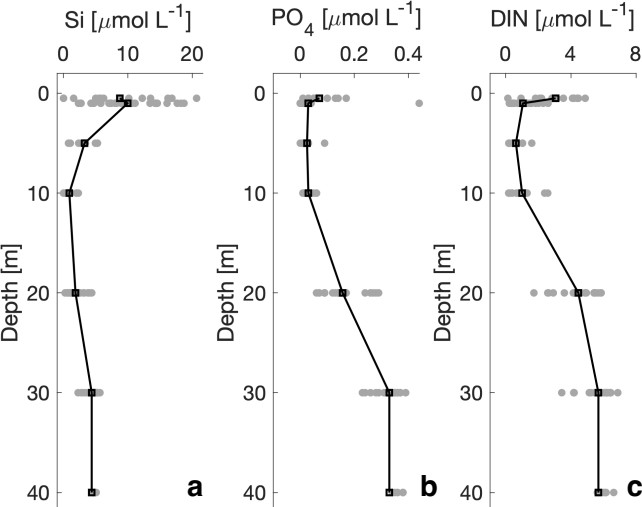

**Figure 9.** Average distributions of macronutrients ($\mu$mol L$^{-1}$) from all stations in the fjord: (a) silicate, (b) phosphate, and (c) dissolved inorganic nitrogen (DIN). The black lines represent the mean for all stations. The grey points show all the data for all stations at each depth.

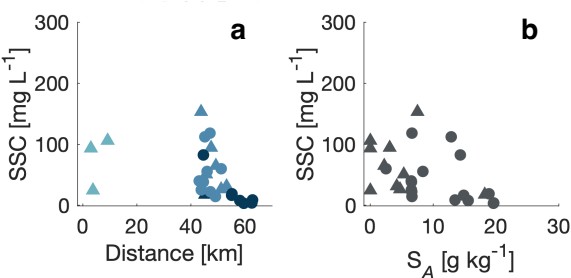

**Figure 10.** Suspended sediment concentration (SSC) versus distance from the glacier (a), and (b) the surface salinity. Data points with salinities in the intervals: <1, 1-13 and >13 g kg$^{-1}$ are shown with colors, where lower salinity is shown in lighter blue and higher salinity in darker blue. Samples taken at the surface (triangles) and at 1 m depth (circles) are indicated.

in the Umivit river. The low concentration further out in the fjord was in general accordance with observations in fjords and coastal waters along west Greenland. Hopwood et al. (2016) observed relatively high concentrations (13 $\mu$g L$^{-1}$) in low-saline

water near a glacier and river outlets in Godthåbsfjord ($\sim$ 65°N) and lower values of $\sim$2 $\mu$g L$^{-1}$ near the fjord mouth. van Genuchten et al. (2021) observed similarly high dFe-values (13 $\mu$g L$^{-1}$) within 10 km of a river outlet in Ameralik fjord (a neighboring fjord to Godthåbsfjord) while their observations around Disko Island ($\sim$ 69°N), i.e., an area close to coastal water masses, showed significantly lower values of $\sim$0.3 $\mu$g L$^{-1}$. However, these values are still signifantly larger than typical open ocean concentrations of typically less than 1 nM (= 0.056 $\mu$g L$^{-1}$) (Boyd, 2002; Boyd and Ellwood, 2010). In summary, the

dFe-concentrations were relatively high in the river water (> 100 $\mu$g L$^{-1}$) and showed a signfiant decrease from more than



100 $\mu$g L$^{-1}$ to less than the detection limit at 5 $\mu$g L$^{-1}$ in the inner fjord and near the river outlet. The high values were closely related to the amount of river water in the surface layer.

Distributions of dCu and dCo also showed a gradual increase along the river whereas Mn, Ni and Zn had an initial low value at the glacier and a relatively constant level from close to the glacier to the river outlet. A similar relation, as found for dFe
between concentration and freshwater content, was seen for dMn, dZn, dCo and dCu (Fig. 6) whereas Ni showed a relatively constant concentration in the inner part of the fjord of ~1 $\mu$g L$^{-1}$. Thus, high concentrations in the Umivit river may also explain the increased concentration of dMn, dZn, dCo and dCu off the Umivit fjord-arm. Comparison between these tracers and concentrations in coastal and open waters around Greenland, showed that the concentrations in the outer part of the transect were relatively high. Campbell and Yeats (1982) measured dMn, dNi and dCu on the shelf off Kangerlussuaq fjord (67.75 °N,
57.08°W) and values near the surface (10 m depth) were 4.4, 0.7 and 1.0 $\mu$g L$^{-1}$, respectively. Thus, concentrations of Mn was significantly higher in the fjord (4.4 $\mu$g L$^{-1}$, Table 1), and also dNi and dCu showed concentrations in the fjord that were above values observed above the shelf. Similarly, the dZn levels observed in this study were up to an order of magnitude higher than observed in the Baffin Bay (Colombo et al., 2019). The dZn concentrations warrant attention due to its role in enzymatic degradation of polysaccharides and its influence on dissolved organic matter (DOM) cycling (Helbert, 2017). Hence, river and
fjord concentrations of dMn, dNi, dCu and dZn all indicated that the fjord was a source of the trace metals to coastal water masses. Distribution of dMo, dV and dU showed low concentration in river water near the glacier. All meausurement of dU were below the detection limit in the river and the highest concentrations were observed at the high-saline outer stations in the fjord transect. There was no, or below detection limit levels of, dU in the surface water near the innermost stations with freshwater content above 2.6 m (Fig. 7e,f). This indicated that the source of dU originated further out in the fjord or from
coastal water masses transported with bottom water into the fjord. The distribution of dV varied both in the river and in the fjord, however, the outermost stations showed values below the detection limit and this was also the case for dMo.

## 4.2   Nutrients and biological production

The river distributions of macronutrients silicate, phosphate, and DIN all showed a general increase along the river. Thus, additional sources of nutrients are present between the glacier and the fjord, likely due to runoff from lakes and possibly
remineralization of organic matter. The concentrations of DIN and phosphate in the river were higher than the surface concentrations in the fjord, which showed that the transport of these macronutrients in the river is important for the cycling and biological uptake in the inner part of the fjord. The vertical distribution of DIN and phosphate showed relatively low values in the upper 10 m and this could also be explained by biological uptake since the breakup of sea ice. The surface silicate concentrations in the fjord were significantly higher than in the river, suggesting an internal silicate source, likely associated
with the weathering of GRF suspended in the river water. Silicate also showed a minimum between the surface and 30 m depth and this could be explained by biological uptake, e.g., uptake by diatoms.

Trace metals also play a crucial role in supporting primary production by acting as essential co-factors in enzymatic and photosynthetic processes. For example, manganese and nickel are vital for metabolic pathways, including carbon fixation and





nitrogen cycling (Hansel et al., 2017; Kuypers et al., 2018). The increased availability of these metals in the inner regions of
the fjord could enhance microbial and phytoplankton activity, particularly during periods of high meltwater discharge.

### 4.3    Future perspectives

Understanding sinks in this saline region of the fjord is crucial, as a significant portion of the dissolved inorganic compounds
are removed from the water column. This study offers insights into the pathways and processes that regulate trace metal and
nutrient dynamics in the transition zone in glacier-influenced fjord systems. Furthermore, the variability in meltwater discharge
and sediment plumes over time highlights the importance of continuous monitoring to capture seasonal and interannual trends.
Further research incorporating the biological response to trace metal inputs from glacial discharge would help clarify the
broader ecosystem impacts, particularly as glacial melt accelerates. Estimating the concentrations of trace metals entering
the ocean from glacial discharge remains challenging and this study uniquely covering the transition from glacier to fjord in
Greenland underlines the importance of the transition zone and the associated sinks that must be considered when modeling or
extrapolating riverine transport into the oceans.

*Data availability.* All trace metal data collected for this study are included in this manuscript.

*Author contributions.* CRV collected and prepared trace metal and nutrient data. CRV, JB and RDD analysed data and computed figures.
RDD collected and measured sediment data. CRV wrote the first draft of the manuscript. All authors revised and contributed to the final
version of the manuscript.

*Competing interests.* The authors declare no competing interests.

*Acknowledgements.* This research was supported by the NOVO Nordisk foundation (NNF22SA0079616). The Carlsberg Foundation also
provided equipment grants (CF19-0051 and CF14-0100). We would like to thank for the helpful assistance from Kangerlussuaq International
Science Support (KISS) and Polar Trophy Hunt. The sattellite image was obtained from the European Space Agency (ESA) Copnernicus
Sentinel-2 L2A.



**Table A1.** Water sampling locations from the GROFS1 field campaign. Trace metals were collected at stations 6-35 (excluding st. 12 and 29). Time is shown as western Greenland time (WGT). Depths from the echo sounder resolved depths down to ∼200 m and at some deeper stations the depth was not resolved. Distances are measured from the glacier. Area type is defined in Table 1. Surface salinity ($S_A$) was bin-averaged between 0.5 - 1.0 m depth, and the freshwater content ($F_W$) was calculated in the upper 15 m.

| Station | Lat (°N) | Lon (°W) | Date (ddmmyy) | Time (WGT) | Depth (m) | Distance (km) | Type - | Surface $S_A$ (g kg$^{-1}$) | $F_W$ (m) |
|---|---|---|---|---|---|---|---|---|---|
| 1 | 66.946 | -50.941 | 21-06-24 | 10:30 | 90 | 45.23 | Fjord | 18.21 | 1.43 |
| 2 | 66.891 | -51.121 | 22-06-23 | 11:00 | 130 | 55.12 | Fjord | 18.89 | 1.32 |
| 3 | 66.862 | -51.273 | 23-06-23 | 11:15 | 225 | 62.46 | Fjord | 19.64 | 1.15 |
| 4 | 66.883 | -51.187 | 24-06-23 | 12:50 | 210 | 58.03 | Fjord | 15.57 | 1.32 |
| 5 | 66.896 | -51.126 | 26-06-23 | 09:50 | 195 | 55.01 | Fjord | 14.86 | 2.06 |
| 6 | 66.879 | -51.220 | 26-06-23 | 13:05 | 214 | 59.5 | Fjord | 19.59 | 1.41 |
| 7 | 66.862 | -51.280 | 27-06-23 | 09:55 | 223 | 62.72 | Fjord | 13.48 | 1.51 |
| 8 | 66.938 | -50.976 | 28-06-23 | 10:40 | 97 | 47 | Fjord | 6.62 | 2.61 |
| 9 | 66.945 | -50.958 | 28-06-23 | 13:20 | >200 | 45.93 | Fjord | 8.36 | 2.28 |
| 10 | 66.947 | -50.939 | 28-06-23 | 14:15 | >200 | 45.1 | Fjord | 12.87 | 2.09 |
| 11 | 66.944 | -50.920 | 28-06-23 | 15:00 | 92 | 44.56 | Fjord | 14.32 | 1.84 |
| 13 | 66.947 | -50.883 | 29-06-23 | 12:23 | 108 | 43.02 | Fjord | 6.51 | 2.90 |
| 14 | 66.946 | -50.902 | 29-06-23 | 14:15 | >200 | 43.78 | Fjord | 6.53 | 2.83 |
| 15 | 66.945 | -50.921 | 29-06-23 | 14:28 | >200 | 44.54 | Fjord | 6.54 | 2.75 |
| 16 | 66.944 | -50.943 | 29-06-23 | 14:40 | >200 | 45.42 | Fjord | 6.56 | 2.68 |
| 17 | 66.943 | -50.960 | 29-06-23 | 14:50 | >200 | 46.11 | Fjord | 6.57 | 2.60 |
| 18 | 66.938 | -50.980 | 29-06-23 | 15:05 | >200 | 47.15 | Fjord | 6.59 | 2.53 |
| 19 | 66.931 | -50.996 | 29-06-23 | 15:17 | >200 | 48.14 | Fjord | 6.60 | 2.45 |
| 20 | 66.925 | -51.010 | 29-06-23 | 15:34 | >200 | 49.01 | Fjord | 6.62 | 2.38 |
| 21 | 66.912 | -51.047 | 30-06-23 | 09:40 | >200 | 51.14 | Fjord | 2.49 | 2.56 |
| 22 | 66.948 | -50.903 | 01-07-23 | 10:45 | >200 | 43.7 | Fjord | 7.50 | 2.19 |
| 23 | 66.943 | -50.946 | 01-07-23 | 11:25 | >200 | 45.59 | Fjord | 5.32 | 1.96 |
| 24 | 66.936 | -50.984 | 01-07-23 | 11:42 | >200 | 47.41 | Fjord | 3.08 | 1.67 |
| 25 | 66.926 | -51.016 | 01-07-23 | 11:55 | >200 | 49.18 | Fjord | 2.21 | 1.92 |
| 26 | 66.915 | -51.051 | 01-07-23 | 12:13 | >200 | 51.12 | Fjord | 4.72 | 1.57 |
| 27 | 66.904 | -51.088 | 01-07-23 | 12:30 | >200 | 53.13 | Fjord | 4.20 | 2.47 |
| 28 | 67.105 | -50.216 | 02-07-23 | 13:45 | 0 | 9.17 | River | 0 | - |
| 29 | 67.151 | -50.038 | 02-07-23 | 15:15 | 0 | 0.09 | River | 0 | - |
| 30 | 67.151 | -50.040 | 02-07-23 | 16:00 | 0 | 0 | Glacier | 0 | - |
| 31 | 67.147 | -50.108 | 02-07-23 | 16:30 | 0 | 2.97 | River | 0 | - |
| 32 | 67.143 | -50.124 | 02-07-23 | 16:42 | 0 | 3.74 | River | 0 | - |
| 33 | 67.006 | -50.678 | 03-07-23 | 01:00 | 0 | 31.99 | River delta | 0 | - |
| 34 | 67.064 | -50.376 | 04-07-23 | 12:41 | 0 | 17.52 | River | 0 | - |
| 35 | 67.028 | -50.584 | 04-07-23 | 13:46 | 0 | 27.23 | River | 0 | - |





**Table A2.** Trace metal data ($\mu$g L$^{-1}$) collected during the GROFS1 campaign. Dissolved trace metals included are: iron (Fe), manganese (Mn), cobalt (Co), copper (Cu), zinc (Zn), nickel (Ni), molybdenum (Mo), arsenic (As), vanadium (V), and uranium (U). Values below detection limit ($\mu$g L$^{-1}$) are shown as: '< detection limit value', for each element respectively. Stations where no trace metal samples were collected are shown as '-'. Chromium (Cr) and lead (Pb) presented mostly all values below detection limit except for station 17 and 35, where Cr was 0.52 and 3.25 $\mu$g L$^{-1}$, respectively. Pb was 0.10 $\mu$g L$^{-1}$ at station 17.

| Station | Lat (°N) | Lon (°W) | Fe | Mn | Co | Cu | Zn | Ni | Mo | As | V | U |
|---|---|---|---|---|---|---|---|---|---|---|---|---|
| 1 | 66.946 | -50.941 | - | - | - | - | - | - | - | - | - | - |
| 2 | 66.891 | -51.121 | - | - | - | - | - | - | - | - | - | - |
| 3 | 66.862 | -51.273 | - | - | - | - | - | - | - | - | - | - |
| 4 | 66.883 | -51.187 | - | - | - | - | - | - | - | - | - | - |
| 5 | 66.896 | -51.126 | - | - | - | - | - | - | - | - | - | - |
| 6 | 66.879 | -51.220 | 5.11 | 5.10 | <0.10 | 1.00 | 1.68 | 0.75 | 2.83 | 0.59 | <0.50 | 0.12 |
| 7 | 66.862 | -51.280 | <5.00 | 3.75 | <0.10 | 0.93 | 9.29 | 0.78 | 3.30 | 0.52 | <0.50 | 0.65 |
| 8 | 66.938 | -50.976 | 17.54 | 5.97 | 0.11 | 1.87 | 93.97 | 0.76 | 1.51 | <0.50 | 0.51 | 0.25 |
| 9 | 66.945 | -50.958 | <50 | 4.96 | <0.10 | 1.02 | 18.58 | 0.69 | 2.01 | <0.50 | <0.50 | 0.46 |
| 10 | 66.947 | -50.939 | <5.00 | 4.28 | <0.1 | 1.12 | 14.78 | 0.80 | 2.91 | <0.50 | <0.50 | 0.64 |
| 11 | 66.944 | -50.920 | <5.00 | 3.89 | <0.10 | 1.11 | 7.29 | 0.71 | 3.26 | <0.50 | <0.50 | 0.53 |
| 13 | 66.947 | -50.883 | 16.08 | 8.06 | 0.12 | 1.21 | 52.95 | 0.62 | 0.57 | <0.50 | 0.65 | <0.10 |
| 14 | 66.946 | -50.902 | 40.93 | 9.87 | 0.16 | 2.49 | 35.17 | 0.78 | 0.37 | <0.50 | 0.60 | <0.10 |
| 15 | 66.945 | -50.921 | 75.2 | 12.47 | 0.16 | 3.12 | 18.68 | 0.84 | 0.22 | <0.50 | 0.68 | <0.10 |
| 16 | 66.944 | -50.943 | 94.77 | 12.88 | 0.22 | 4.12 | 31.25 | 0.78 | 0.22 | <0.50 | 0.69 | <0.10 |
| 17 | 66.943 | -50.960 | 75.24 | 10.26 | 0.18 | 3.45 | 60.31 | 0.93 | 0.34 | <0.50 | 0.76 | <0.10 |
| 18 | 66.938 | -50.980 | 13.67 | 6.63 | <0.10 | 1.20 | 121.78 | 0.64 | 0.65 | <0.50 | 0.79 | <0.10 |
| 19 | 66.931 | -50.996 | <5.00 | 4.13 | <0.10 | 0.83 | 11.56 | 0.91 | 1.98 | <0.50 | <0.50 | 0.43 |
| 20 | 66.925 | -51.010 | <5.00 | 4.06 | <0.10 | 0.99 | 34.29 | 0.92 | 2.38 | <0.50 | <0.50 | 0.49 |
| 21 | 66.912 | -51.047 | <5.00 | 7.63 | <0.10 | 0.76 | 5.43 | 0.61 | 0.70 | <0.50 | 0.59 | <0.10 |
| 22 | 66.948 | -50.903 | 8.97 | 8.43 | <0.10 | 0.95 | 26.31 | 0.79 | 0.37 | <0.50 | 0.58 | <0.10 |
| 23 | 66.943 | -50.946 | <5.00 | 7.72 | 0.11 | 0.77 | 3.73 | 0.73 | 0.65 | <0.50 | 0.64 | <0.10 |
| 24 | 66.936 | -50.984 | <5.00 | 7.84 | <0.10 | 0.68 | 2.47 | 0.62 | 0.58 | <0.50 | 0.81 | <0.10 |
| 25 | 66.926 | -51.016 | <5.00 | 7.60 | <0.10 | 0.72 | 8.90 | 0.62 | 0.55 | <0.50 | 0.66 | <0.10 |
| 26 | 66.915 | -51.051 | <5.00 | 6.80 | <0.10 | 0.78 | 21.95 | 0.71 | 0.81 | <0.50 | 0.67 | 0.14 |
| 27 | 66.904 | -51.088 | <5.00 | 5.53 | <0.10 | 0.77 | 5.65 | 0.90 | 1.33 | <0.50 | <0.50 | 0.24 |
| 28 | 67.105 | -50.216 | 16.02 | 9.20 | <0.10 | 1.29 | 27.9 | 0.65 | <0.10 | <0.50 | <0.50 | <0.10 |
| 29 | 67.151 | -50.038 | - | - | - | - | - | - | - | - | - | - |
| 30 | 67.151 | -50.040 | 12.50 | 2.92 | <0.10 | 0.58 | 3.96 | 0.40 | <0.10 | <0.50 | <0.50 | <0.10 |
| 31 | 67.147 | -50.108 | 22.20 | 10.21 | 0.12 | 1.09 | 37.38 | 0.60 | <0.10 | <0.50 | 0.51 | <0.10 |
| 32 | 67.143 | -50.124 | 25.04 | 3.05 | <0.10 | 2.56 | 34.93 | 1.54 | <0.10 | <0.50 | <0.50 | <0.10 |
| 33 | 67.006 | -50.678 | 37.16 | 9.33 | <0.10 | 1.58 | 27.43 | 0.46 | 0.27 | <0.50 | 1.00 | <0.10 |
| 34 | 67.063 | -50.376 | 81.32 | 9.04 | 0.14 | 2.23 | 48.15 | 0.69 | 0.19 | <0.50 | 0.77 | <0.10 |
| 35 | 67.028 | -50.584 | 128.94 | 10.26 | 0.18 | 3.1 | 18.52 | 0.87 | 0.20 | <0.50 | 0.78 | <0.10 |



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
