# Peer review of "Trace metal distributions in the transition zone from the Greenland Ice-Sheet to the surface water in Kangerlussuaq fjord (67 $^{\circ}$ N)"

_EGUsphere, 2025_

## Referee Comment (RC1)

Many thanks for the opportunity to review "Trace metal distributions in the transition zone from the Greenland Ice-Sheet to the surface water in Kangerlussuaq fjord (67N)". The aim of this paper is to present new nutrient and trace metal data from this West Greenland fjord, in the context of the impact of glacial meltwaters and sediments on the supply of important elements to the ocean. More datasets and discussion surrounding this important topic are welcome in the literature and will be of great interest to your readership. I found this paper very clear and enjoyable to read. I only have a few main suggestions for clarifying the methods section, and a few additional minor comments to be addressed.

Main comments:

I would like to see more details regarding the trace metal and nutrient analyses in the methods section.

Line 138 states that the samples were calibrated against a "pre-made solution". I think the authors need to give more details about this solution (commercial? In-house? What was its composition?). In particular, matrix matching is very important for ICP measurements (i.e., making sure that calibration standards have the same composition, as close as possible, to samples). This means that the calibration standards would need to be made up to an appropriate salinity using artificial seawater (and different standards may well be needed for river vs fjord samples). It would be good to verify if the authors have considered such matrix effects (at least show that they do not impact the accuracy and precision of the final measurements).

Line 138 states that "certified standards" were measured. Please include details of, and measurements of, these reference materials as this will help determine the accuracy (and precision) of the sample analyses.

Line 137 states that each sample was measured three times – what do these replicates indicate about analytical precision?

Line 138 says that "Background levels" were analysed, but what do these background levels represent? MilliQ water that was processed the same way as the samples (i.e., a laboratory blank) or an instrumental blank?

Section 2.4. Were reference materials run for the macronutrient samples? If so, please report these data as well. If not, this needs to be stated. What is the precision of the measurements in each case?

Note that the data table in the appendix (Table A2) does not show the individual measurements or the uncertainties associated with each measurement. I would either put in the data for each replicate or (probably more straightforwardly) include a standard deviation or standard error.

Apologies if I've missed it, but I can't find the original macronutrient data in the paper submission portal. I would suggest including these too, or at least a link to a published and openly accessible dataset.

My general suggestion would be to make these datasets available in a usable format i.e., .csv format or similar. (I would recommend an external data repository to do this for full accessibility).

Minor comments:

Line 34: I found the addition of the two sentences on toxic metals a little out of place. There is more literature out there on toxic metals and glacial weathering that are not referenced here, and two sentences doesn't really do the topic "justice". Given that these elements were not discussed in the manuscript, I would suggest taking these sentences out.

Line 49: The authors rightly point out that there is iron limitation in regions of the North Atlantic in summer, but it might also be interesting (and useful for the paper) to mention that there is evidence also for season silicon limitation of diatom production in this region and elsewhere in the Arctic [1-3].

Line 124: Some of this methods section (e.g., "surface salinity showed a minor change along the transect...") fits better in the results section. I would suggest the authors check through their methods and move any results to the appropriate section.

Line 205: I agree with the authors that the observation that the highest concentrations of many of the trace metals were where the freshwater content was highest, indicating an "impact from runoff". Do the authors mean glacial runoff specifically, or all types of runoff? As stated in the methods section, Fw also reflects freshwater from precipitation and sea-ice melt. Could these other sources be complicating the picture here? Or can the authors argue that Fw is dominated by glacial runoff? (which I suspect it probably is!). It might be worth just clarifying that here.

Line 284: I agree that the macronutrient data points towards additional sources due to lake runoff or in situ dissolution, but I would suggest expanding "remineralization of organic matter" to include "remineralization of organic matter and GRF", or similar, to be consistent with Lines 289-290.

Section 4.2. could be expanded. I would suggest to the authors to include some brief comparisons with published findings on macronutrients in Greenlandic fjords (similar to that in Section 4.1).

References:

1    Krause, J. W. *et al.* Biogenic silica production and diatom dynamics in the Svalbard region during spring. *Biogeosciences* **15**, 6503-6517 (2018).
2    Krause, J. W. *et al.* Silicic acid limitation drives bloom termination and potential carbon sequestration in an Arctic bloom. *Scientific Reports* **9**, 8149 (2019).
3    Ng, H. C. *et al.* Detrital input sustains diatom production off a glaciated Arctic coast. *Geophysical Research Letters* **51**, e2024GL108324 (2024).

---

## Author Comment (AC1)

**Response to reviewer document**

We sincerely appreciate the feedback from the reviewer, which has improved the quality of our manuscript. Following the reviewer's suggestions, we have included a more detailed explanation of the methods (including reference materials, and quality checks) for the trace metal analysis. We have also expanded the discussion of nutrients in the results section. We believe these revisions have improved the manuscript.

**Reviewer 1**

Main comments:
I would like to see more details regarding the trace metal and nutrient analyses in the methods section.

Line 138 states that the samples were calibrated against a "pre-made solution". I think the authors need to give more details about this solution (commercial? In-house? What was its composition?).

Thank you for the comments on the trace metal methodology section. We agree with the reviewer that adding a more detailed description of the methods will ensure replicability and transparency for the reader. We have added a more detailed description of the laboratory methods used to analyse the trace metal samples as follows (Line 137):

**The elemental concentration of trace metals was measured on Perkin Elmer Avio 200 Inductive Coupled Plasma-Optical Emission Spectrometer (ICP-OES) at the Sustain Lab (Danish Technical University, Denmark). Based on repeated measurement of certified in-house standards (SCP Science EnviroMAT), the relative standard deviation (RSD) of the measurements was calculated.**

In particular, matrix matching is very important for ICP measurements (i.e., making sure that calibration standards have the same composition, as close as possible, to samples). This means that the calibration standards would need to be made up to an appropriate salinity using artificial seawater (and different standards may well be needed for river vs fjord samples). It would be good to verify if the authors have considered such matrix effects (at least show that they do not impact the accuracy and precision of the final measurements).

**The sea water samples were diluted 10 times to decrease the salinity, and the calibration curves and standards were prepared in a corresponding matrix solution made with artificial pure NaCl.** This has been added following the text from the comment above, on line 143.

Line 138 states that "certified standards" were measured. Please include details of, and measurements of, these reference materials as this will help determine the accuracy (and precision) of the sample analyses.

The standard used for any water samples was EnviroMAT. We added this in the paragraph answering the comment on line 138, as follows:

**Based on repeated measurement of certified in-house standards (SCP Science EnviroMAT), the relative standard deviation (RSD) of the measurements was calculated.**

Line 137 states that each sample was measured three times – what do these replicates indicate about analytical precision?

Each sample injection was analyzed 3 times, in order to estimate the RSD of each individual measurement. This has been added in line 140, following the answer to the comment below about line 138, as follows:

**Furthermore, each injection of the sample was measured three times, in order to estimate the RSD of each individual measurement. The method detection limit (MDL) was calculated from the calibration curve. To enhance the measurement precision (lowest point ~0.05 mg/L), axial view setting was used for measurement of concentrations <1mg/L and radial view for concentrations >1mg/L. Processing of the data was carried out in the Syngistic™ for ICP Software v. 2.0 from Perkin Elmer.**

Line 138 says that "Background levels" were analysed, but what do these background levels represent? MilliQ water that was processed the same way as the samples (i.e., a laboratory blank) or an instrumental blank?

**The background level from laboratory blanks were analyzed and included in the corrections and detection limit calculations.** The concentration in blanks is normally lower than the lowest calibration point. This is included in line 144.

Section 2.4. Were reference materials run for the macronutrient samples? If so, please report these data as well. If not, this needs to be stated. What is the precision of the measurements in each case?
The information of reference materials and reagents for nutrient samples are described in the cited reference (Grashof, 1983, and Grashof et al., 2009).

We provide detection limits for all the nutrient in section 2.4. The nutrient ranges are well above the detection limits.

Note that the data table in the appendix (Table A2) does not show the individual measurements or the uncertainties associated with each measurement. I would either put in the data for each replicate (probably more straightforwardly) or include a standard deviation or standard error.

Thank you for this suggestion. We agree with the reviewer that using standard deviation or standard error will be more helpful for the reader. Following this suggestion, we have updated table A2 and included all replicates for transparency.

Apologies if I've missed it, but I can't find the original macronutrient data in the paper submission portal. I would suggest including these too, or at least a link to a published and openly accessible dataset. My general suggestion would be to make these datasets available in a usable format i.e., .csv format or similar. (I would recommend an external data repository to do this for full accessibility).

Thank you for this comment. The nutrient data is going to be published in the public database of ICES as: Nutrient and CTD-data are archived at ICES (http://www.ices.dk/). We have included this in the Data availability statement. All the trace metal data are presented in themanuscript.

**Minor comments:**
Line 34: I found the addition of the two sentences on toxic metals a little out of place. There is more literature out there on toxic metals and glacial weathering that are not referenced here, and two sentences doesn't really do the topic "justice". Given that these elements were not discussed in the manuscript, I would suggest taking these sentences out.
 We have deleted these sentences.

Line 49: The authors rightly point out that there is iron limitation in regions of the North Atlantic in summer, but it might also be interesting (and useful for the paper) to mention that there is evidence also for season silicon limitation of diatom production in this region and elsewhere in the Arctic 1-3 .
Thank you for pointing this out, we definitely agree with this comment. We have added a mention of the Si limitation based on the references provided, as follows (Line 42):

**Silicon limitation of diatom production is also present in this region and in other parts of the Arctic (Krause et al., 2018, 2019; Ng et al., 2024). Hence, alongside macronutrients such as phosphate and nitrate, levels of silicate and trace metals regulate oceanic biological production in this region.**

Line 124: Some of this methods section (e.g., "surface salinity showed a minor change along the transect…") fits better in the results section. I would suggest the authors check through their methods and move any results to the appropriate section.

We removed the following sentences from the methods section 2.2. and moved to the results section 3.1.

Line 160: **Thus, surface salinity showed a minor change along the transect (Fig. 3c).**

Line 174: **The change in Fw of ~0.5 m reflected that the depth of the surface plume decreased along the transect into the fjord (Fig. 3b, Table A1).**

Line 205: I agree with the authors that the observation that the highest concentrations of many of the trace metals were where the freshwater content was highest, indicating an "impact from runoff". Do the authors mean glacial runoff specifically, or all types of runoff? As stated in the methods section, Fw also reflects freshwater from precipitation and sea-ice melt. Could these other sources be complicating the picture here? Or can the authors argue that Fw is dominated by glacial runoff? (which I suspect it probably is!). It might be worth just clarifying that here.

This is mainly from glacial run off. Sea ice melt and precipitation can not explain the large amount of freshwater in the inner part of the fjord, as described in the added text below. We have clarified this in the sentence as follows (Line 206):

**The gradient in Fw is dominated by river runoff. The impact of sea ice melt would approximately be equal along the fjord and of the order of 1 m under the assumption of a typical sea ice thickness of ~1 m in the fjord. The precipitation on the surface makes a small contribution as this period was relatively dry (section 2.1). Thus, the freshwater gradient decreasing from ~3 to ~1 meter at the outer stations mainly represents river water.**

Line 284: I agree that the macronutrient data points towards additional sources due to lake runoff or in situ dissolution, but I would suggest expanding "remineralization of organic matter" to include "remineralization of organic matter and GRF", or similar, to be consistent with Lines 289-290.

Thank you for this suggestion. We have added "and GRF" to the sentence as follows:

**Thus, additional sources of nutrients are present between the glacier and the fjord, likely due to runoff from lakes and possibly remineralization of organic matter and GRF.**

Section 4.2. could be expanded. I would suggest to the authors to include some brief comparisons with published findings on macronutrients in Greenlandic fjords (similar to that in Section 4.1).

We acknowledge the reviewer's suggestion about adding a more detailed discussion of the macronutrient distributions. This is also the focus of an ongoing manuscript and analysis of macronutrients and phytoplankton in the fjord. The focus of this study is on the trace metal distribution and therefore we have decided to keep the discussion of nutrient distributions as it is.

---

## Author Comment (AC2)

**Response to reviewers document**

**Reviewer 2**

Thank you for the invitation to review the paper "Trace metal distributions in the transition zone from the Greenland Ice-Sheet to the surface water in Kangerlussuaq fjord (67 ◦N)". The manuscript outlines the spatial distribution of trace elements, macronutrients, and suspended sediments from the land-terminating Russell Glacier, through the Akuliarusiarsuup Kuua, to the Kangerlussuaq fjord in western Greenland. It indicates potential sources for these elements and nutrients as well as discusses potential sinks and their relevance for biological productivity in a changing climate. This study provides comprehensive sampling of the near-glacial fjord, capturing the salinity transition well – this is quite novel. Although the manuscript is largely well-written, some improvements need to be made, particularly with regards to the methods and discussion sections, and the interpretation of data. The discussion section in particular could do with additional exploration and explanation of the dataset.

We sincerely appreciate the reviewer's thoughtful feedback. In line with their suggestions, we have revised the manuscript to include a more detailed description of the trace metal analysis methods. Additionally, we have expanded the background and discussion sections and incorporated several new references to strengthen the context. We believe these revisions have significantly improved the manuscript and are grateful to the reviewer for their time and constructive input.

**Major comments:**

The methods section should be reworked to include more details on both the field and analytical methods. It was a bit unclear how the trace metal samples were collected, what is the "trace- metal clean bottle equipment" (Line 128)? I assume the Niskin bottle was not a trace-metal clean bottle and so it was not used for trace element collection at depth, although I think it would have been an interesting addition to the study. Please clarify these field methods a bit, as it reads a bit confusing moving between fjord and river samples and how these were each collected.

I agree with the previous anonymous reviewer that some more details should also be included on the ICP-OES analysis methods. How were the three replicates factored into error analysis? I also agree that it is important to include information on how calibration standards were made, what the "pre-made solutions" and "certified standards" were, and how these allow you to be sure your results are accurate, given the low concentrations of some of these elements for ICP-OES analysis and the matrix differences between fjord and river samples. Were fjord samples diluted or undergo any extra prep (Lines 128-139)? Did you use any certified reference material?

We have expanded the section for the trace metal methods. We include the final section below, and more specific answers to your questions after that. Please note that we originally stated that the analyses were carried out with an ICP-OES, however, we corrected that to

"ICP-MS". Elements analysed using an ICP-OES included Na, Mg, K, Ca, Al, and Be. However these results are not included in this manuscript.

(Line 129) **Trace metal samples were collected at 28 stations at the surface, with trace-metal clean low-density polyethylene (LDPE) bottles. Samples were collected from undisturbed water in the fjord and in the river. One station (st. 33) was located beside a bridge at the entrance of the river delta and before the discharge from the small town of Kangerlussuaq. Samples in the river were collected facing the current to avoid contamination. All equipment used to sample the concentration of trace metals in the water was prepared following the GEOTRACES protocol (Cutter et al., 2010). Briefly, LDPE bottles were washed in Decon for two weeks before being placed in an acid bath (HCl 6M) for an additional four weeks. All bottles were then rinsed three times with ultra-pure MilliQ water and triple bagged for transportation. To measure dissolved concentrations, aliquots were syringe-filtered through acid-washed Pall Acropak Supor capsule filters (0.2 µm). Prior to analysis the samples were acidified to 2% $HNO_3$ and 0.5 % HCl (v/v). The samples were analyzed by ICP-MS (7850x, Agilent Technologies) with Yttrium as the internal standard at the Sustain Lab (Danish Technical University, Denmark). The instrument was equipped with Platinum tipped skimmer and sample cones, a double pass Scott spray chamber operated at 2 °C and a Micromist nebulizer. Elements analyzed in He collision mode were with a Helium flow of 5 ml min$^{-1}$.**

**Based on repeated measurement of certified in-house standards (SCP Science EnviroMAT), the relative standard deviation (RSD) of the measurements was calculated. Furthermore, each injection of the sample was measured three times, in order to estimate the RSD of each individual measurement. The method detection limit (MDL) was calculated from the calibration curve. To enhance the measurement precision (lowest point 0.05 mgL$^{-1}$), axial view setting was used for measurement of concentrations <1mg L$^{-1}$ and radial view for concentrations >1mg L$^{-1}$. The sea water samples were diluted 10 times to decrease the salinity, and the calibration curves and standards were prepared in a corresponding matrix solution made with artificial pure NaCl. The background level from laboratory blanks were analyzed and included in the corrections and detection limit calculations. Quantification limits for each element are listed in Table 1. Processing of the data was carried out in the SyngisticTM for ICP Software v. 2.0 from Perkin Elmer.**

More specific answers to your questions, which aligned with comments from reviewer 1, on the trace metal analyses below:

*Response to reviewer 1 (copy to reviewer 2).*

What are the samples calibrated against? A pre-made solution that is commercial? Made in-house? What is its composition?

Calibration standards are always commercial single element solutions or commercial pre-made mixes. For ICP-MS measurements, pre-made commercial mixes are used, such as the

Quality Control Standard TruQms, 5% JNO3/trace Tartaric Acid/trace HF, 125ml from PerkinElmer (https://www.perkinelmer.com/uk/product/quality-control-standard-21-125-ml-n9303837). These are diluted in-house to the appropriate concentrations for the calibrations.

Can you provide some information on how calibration standards were made, what the "pre-made solutions" and "certified standards" were, and how these allow you to be sure your results are accurate, given the low concentrations of some of these elements for ICP-OES analysis and the matrix differences between fjord and river samples. Were the calibration standards up to an appropriate salinity using artificial seawater? and were different standards used for river vs fjord samples? Or in other words, was the salinity checked for each sample before applying an appropriate calibrated solution?

Our standard procedure when measuring sea water was to matrix match the calibrations standards with 3% NaCl (TraceMetal grade). Since all samples were analysed with internal standards (typically Y, Sc and In) a potential matrix interference on the sample introduction will be spotted. For detection by ICP-MS we used He collision cell technology as well, prior to detection, in order to remove any potential poly atomic interferences formed by association primarily with Cl (for example, AgCl interfering on As).

Can you provide details of, and measurements of, the reference materials used? to determine the accuracy (and precision) of the sample analyses.

The reference material used is a certified drinking water standard: EnviroMAT-Drinking Water High from analyticchem (https://www.scpscience.com/en/products/details?id=140-025-032&name=enviromat-drinking-water-high).

We added this in the paragraph answering the comment on line 138, as follows:

**Based on repeated measurement of certified in-house standards (SCP Science EnviroMAT), the relative standard deviation (RSD) of the measurements was calculated.**

The text says that the samples were analysed 3 times. What do these replicates indicate about analytical precision?

Each sample injection was analyzed 3 times, meaning the same sample is injected three times (as it gives instrument variance for each sample), in order to estimate the RSD of each individual measurement. This has been added in line 140, following the answer to the comment below about line 138, as follows:

**Furthermore, each injection of the sample was measured three times, in order to estimate the RSD of each individual measurement. The method detection limit (MDL) was calculated from the calibration curve. To enhance the measurement precision (lowest point ~0.05 mg/L), axial view setting was used for measurement of concentrations <1mg/L and radial view for concentrations >1mg/L.**

**Processing of the data was carried out in the Syngistic™ for ICP Software v. 2.0 from Perkin Elmer.**

This is also included in Table 1 and Table A2.

How were the three replicates factored into error analysis?

The replicates provide information on the reproducibility of the measurements, however, they are not directly factored into an error analysis.

Were the background levels analysed? Were there any blanks? Did you measure the trace metal concentration in those (laboratory) blanks?

Blanks are always included in the run. The concentration in blanks is normally lower than the lowest calibration point. If the blanks were higher than the typical instrument QL, a new QL based on the blanks was calculated. Blanks are always matrix matched. This has been added following the text from the comment above, on line 143-144 as follows:

**The sea water samples were diluted 10 times to decrease the salinity, and the calibration curves and standards were prepared in a corresponding matrix solution made with artificial pure NaCl.**

**The background level from laboratory blanks were analyzed and included in the corrections and detection limit calculations.**

In addition to this, we changed "trace- metal clean bottle equipment" (Line 128), to **trace-metal clean low-density polyethylene (LDPE) bottles,** following the reviewer's comment.

Along with the improvement on the discussion of error analysis, include error bars on your points in your figures and indicate what type of error is plotted in the figure captions. Also include standard deviation or standard errors for your measurements in Table A2. Make sure to also include in the Appendix a table for the macronutrient data.

 We appreciate the reviewer's suggestion regarding the inclusion of error bars and error metrics. In our study, each data point represents a single discrete sample per station as no technical triplicates were taken; thus, no statistical variability (e.g., standard deviation or standard error) can be derived across replicates for trace metals.. As such, adding error bars to the figures would not be meaningful or informative. However, to address the concern about analytical uncertainty, we have (1) updated Table A2 to include all measurements in the sampling area, inc. duplicates at the same station where applicable (attached at the end of this document), and (2) included a new table in the Supplementary Information, Table S1 (also attached at the end of this document), with all the standard deviation (RSD, reference

standard) and the certified standard recovery (recovery of the nominal value, % Rec) for each trace metal element, based on reported analytical precision from the laboratory.

The nutrient data is going to be published in the public database of ICES as: Nutrient and CTD-data are archived at ICES (http://www.ices.dk/). We have included this in the Data availability statement.

We hope this explanation and the revisions sufficiently address the reviewer's comments regarding measurement uncertainties.

I also have some reservations about the interpretation of the data. A major consideration for the purposes of the paper's discussion is that the discharge from the Russell Glacier outlet is much lower than that from Leverett Glacier, which mixes with Russell water at the confluence between your Stations 28 and 34. What is described as in-stream weathering and changes from near- glacier to farther samples is more simply likely the result of mixing between meltwater rivers from these two catchments (an increase in iron of this amount from in-stream weathering would be quite remarkable). This should also be considered for the confluence with the Ørkendalen River just north of Kangerlussuaq.

We agree that the mixing of meltwater from Leverett Glacier and the Ørkendalen River will influence the geochemical composition of the downstream sites. To clarify this, we have revised the discussion to explicitly consider both in-stream weathering and catchment mixing as possible explanations for the observed changes, including the increase in iron concentrations. However, given the limited spatial resolution and lack of hydrological discharge data, we acknowledge that it is not possible to definitively attribute the changes to one process over the other. We now clarify this limitation in the revised manuscript and have taken a more cautious approach in our interpretation of the results. We changed parts of the discussion as follows:

> (Line 248) **The highest value of dFe (129 μg L$^{-1}$) was observed mid-way between the glacier and the river delta, suggesting the presence of significant dFe sources along the river (Table A2). One possible explanation is enhanced weathering and metal mobilization or external sources associated with mixing of meltwaters from different sub-catchments, e.g., contributions from joining rivers and streams between the Russel glacier and the Kangerlussuaq fjord such as the Ørkendalen river.**

Finally, the referencing and general field site background are quite underdeveloped. The authors need to ensure they are referencing appropriate literature at the correct time, as this will help aid the discussion and interpretation of data. Some references are missing (e.g. Martin et al. 2020, Hawkings et al. 2020, Yde et al. 2014), and while some are cited in the introduction they are not utilized fully (see specific comments below).
*1. Macdonald, R. W., and F. A. Mclaughlin (1982) The Effect of Storage by Freezing on Dissolved Inorganic-Phosphate, Nitrate and Reactive Silicate for Samples from Coastal and Estuarine Waters*

*2. Macdonald, R. W., F. A. Mclaughlin, and C. S. Wong (1986) The Storage of Reactive Silicate Samples by Freezing*
*3. Martin et al. (2020) Comparisons of Nutrients Exported From Greenlandic Glacial and Deglaciated Watersheds*
*4. Hawkings et al. (2020) Enhanced trace element mobilization by Earth's ice sheets*
*5. Yde et al. (2014) Meltwater chemistry and solute export from a Greenland Ice Sheet catchment, Watson River, West Greenland*

Thank you for this helpful comment. We have now incorporated key studies on glacial meltwater biogeochemistry and trace metal export that are directly relevant to our work, including Hawkings et al. (2020), Martin et al. (2020), and Yde et al. (2014). Hawkings et al. (2020) is particularly relevant, as it offers detailed insights into glacial iron export from the Greenland Ice Sheet to the ocean. Our study builds on this by providing a more resolved view of trace metal transitions from the glacier, through the meltwater river and estuary, into the fjord. More specific comments below.

**Some more specific line-by-line comments:**

Lines 30-32: Consider removing or rephrasing the description of GRF as "chemically immature", as although the GRF is likely more similar to source rock, there is still the possibility for strong chemical weathering at the bed.

We changed the description of GRF to: (Line 31-32) **This heavy physical erosion makes the GRF relatively less chemically mature compared to more weathered sediments, and its mineralogical composition is therefore very similar to its source rocks.**

Lines 34-37: "Mercury concentrations have been found to be very low in meltwater entering Kangerlussuaq fjord (Jørgensen et al., 2024) in accordance with the analyzed composition 35 of GRF (Sarkar, 2021). Sarkar (2021) also found that other toxic substances in GRF from different locations around Greenland were present in very low concentrations." Remove these sentences given toxic elements (including mercury) are not discussed in the paper and this feels out of context.

This has been removed from the introduction.

Figure 1: I'd recommend renaming your stations in the fjord to make more sense spatially, so they are easier to reference later.

The numbering of stations reflected the time of sampling, and this information is also relevant for understanding the data set. Therefore, we have decided to keep the numbering and labelling of stations as they were.

Figure 2: Indicate how discharge was measured and/or where this data came from.
The reference to the data is given in the figure text (van As, D.: Watson River discharge (2006-2023) daily.txt, Watson river discharge, GEUS Dataverse, V3,

https://doi.org/10.22008/FK2/XEHYCM/2A5USE, 2022) and the link provides both data and further explanation. We included the following in the figure legend:
**The river data is a reanalysis product (van As et al.,2018).**

Lines 94-95: What does "in proximity" to the glacier mean? Are you sampling directly at the subglacial portal or on the ice surface for either of these? If not, I'd recommend just describing all of these samples as meltwater river samples.

We sampled on the ice of the glacier, however, no subglacial water was sampled. As such, we have followed the reviewer's recommendation and changed all samples to "**meltwater river samples**" instead. We edited the text as below:
Line 92: **A total of 27 samples were collected from stations within the fjord. Additionally, 8 meltwater river samples were collected: 6 from stations along the river and 2 from stations located less than 200 meters from the glacier (Table A1).**

Line 135: Likely a typo here for the filter size as 0.2µm would make sense for looking at "dissolved" fraction.
Yes that is a typo, thank you, we have corrected "0.2mm" to 0.2µm

Line 136: What type and make of ICP-OES?
As mentioned earlier in this document, please note that we originally stated that the analyses were carried out with an ICP-OES, however, we corrected that to "ICP-MS". Elements analysed using an ICP-OES included Na, Mg, K, Ca, Al, and Be. However these results are not included in this manuscript.
We have now added the following:

Line 134: **The samples were analyzed by ICP-MS (7850x, Agilent Technologies) with Yttrium as the internal standard at the Sustain Lab (Danish Technical University, Denmark). The instrument was equipped with Platinum tipped skimmer and sample cones, a double pass Scott spray chamber operated at 2 °C and a Micromist nebulizer. Elements analyzed in He collision mode were with a Helium flow of 5 ml/min.**

Table 1: I'd recommend labeling elemental values in the table as "below QL" as opposed to providing the QL as the concentration number. Detection limits are quite high on ICP-OES, and so displaying values in this way can be misleading. Also, for Fe, the average value displayed is below the QL which is confusing and should be addressed.
Thank you, we have now changed the quantification limit value from each trace metal element for "**<QL**". The QL for dFe had a typo and is in fact 1 and not 5. That has been corrected now:

**Table 1.** Mean and standard error for concentrations ($\mu g\ L^{-1}$) of dissolved trace metals. The transect is divided into five distinct areas: glacier, river, river delta, inner fjord (low salinity < 13) and fjord (high salinity). Dissolved trace metals included are: iron (Fe), manganese (Mn), cobalt (Co), copper (Cu), zinc (Zn), nickel (Ni), molybdenum (Mo), arsenic (As), vanadium (V), and uranium (U). Values below quantification limit (QL) are shown as "<QL" and the QL for each element ($\mu g\ L^{-1}$) is shown respectively. A table including all trace metal samples is included in Table A2 and the corresponding instrument uncertainties are shown in Table S1.

| | Glacier (n = 1) | River (n = 5) | River delta (n = 1) | Inner fjord (n = 19) | Fjord (n = 6) | QL |
|---|---|---|---|---|---|---|
| **dFe** | 17.76 ±0 | 47.16±36.92 | 36.67 ±0 | 19.61±27.72 | 3.75±1.21 | 1 |
| **dMn** | 9.4 ±0 | 7.21±4.17 | 9.8 ±0 | 7.60±2.72 | 4.4±1.68 | 0.5 |
| **dCo** | <QL | 0.13±0.03 | <QL | 0.13±0.04 | <QL | 0.1 |
| **dCu** | 1.18 ±0 | 1.56±1.03 | 1.4 ±0 | 1.36±1 | 1±0.14 | 0.5 |
| **dZn** | 22.64 ±0 | 23.26±17.97 | 19.96 ±0 | 30.87± 32.35 | 7.13±5.1 | 0.5 |
| **dNi** | 0.65 ±0 | 0.61±0.34 | <QL | 0.79±0.16 | 0.73±0.05 | 0.1 |
| **dMo** | <QL | 0.14±0.05 | 0.27 ±0 | 1.03±0.77 | 3.07±1.18 | 0.1 |
| **dAs** | <QL | <QL | <QL | <QL | 0.54±0.07 | 0.5 |
| **dV** | <QL | 0.61±0.15 | 0.997 ±0 | 0.61±0.11 | <QL | 0.5 |
| **dU** | <QL | <QL | <QL | 0.21±0.16 | 0.60±0.29 | 0.1 |

Lines 141-142: Was this also how your trace element samples were collected? Recommend then moving this up to start with a general discussion of your field methods followed by analytical methods.

Trace metal samples were not collected using a Niskin bottle, they were collected directly from the surface using LDPE (acid-washed) bottles. Thus, we have decided to leave both (trace metals and nutrients) collection descriptions separate.

Line 144: Frozen samples can lead to loss in molybdate reactive silicate, so it should be mentioned here that values may be underreported (Macdonald et al., 1982, Macdonald et al, 1986).

We acknowledge the effects of freezing macro-nutrient samples. However, please note that we filtered all of our samples through a 0.2 μm filter to remove sediments and to avoid any effects in turbid samples. We added the following in the methods section for nutrients:

Line 151: **While we acknowledge that freezing turbid samples can affect silicate concentration measurements (Macdonald et al., 1982, Macdonald et al., 1986), filtering through a 0.2 μm filter minimizes turbidity-related loss of molybdate-reactive silicate.**

Lines 144-146: Were any reference materials used for these macronutrient determinations?
Reference material and methods follow the cited reference (Grasshoff, 1983).

Line 205: Do you only mean glacial discharge here or other sources of runoff like precipitation, lake runoff, etc.?
Runoff includes all freshwater sources. We added the following:
Line 215: **That implies that these tracers had the highest concentrations at stations with the largest impact from runoff, including all freshwater sources.**

Line 219: Is it possible the units for Si are wrong here? I'd expect Si to be around 5-20 in mol/L.
Thank you. The unit has been corrected to **µmol L$^{-1}$** (also in line 220).

Figure 6 and 7: I found the use of Fw and the unit (m) quite confusing, particularly as m is the SI for meters. This should be clarified and the unit changed.
The unit is correct. The application of freshwater content quantifies the amount of freshwater required to dilute a water column with a specified depth to the observed salinity, as described in the text and further explained in the reference.

Lines 238-240: I would consider the larger river from Leverett glacier as the potential source for this. An increase of almost 10x seems too high to be explained by instream weathering.
From the answer to the main comment, we added this:

(Line 142) **The highest value of dFe (129 µg L$^{-1}$) was observed mid-way between the glacier and the river delta, suggesting the presence of significant dFe sources along the river (Table A2). One possible explanation is enhanced weathering and metal mobilization or external sources associated with mixing of meltwaters from different sub-catchments, e.g., contributions from joining rivers and streams between the glacier and the fjord.**

(Line 255) **This is consistent with recent findings from Hawkings et al. (2020), who reported extremely high concentrations of dFe (up to 20,900 nM, 1,170 µg L$^{-1}$) and large annual fluxes (1.4 Gmol y-1) from the Leverett Glacier subglacial system. Their study highlights the geochemical reactivity and potential for high trace metal export from this catchment, which drains into the Watson River. Importantly, Leverett Glacier is hydrologically connected to the main meltwater river sampled in this study, strengthening the likelihood that it contributes significantly to the elevated dFe values observed mid-river.**

Lines 241-245: Yde et al. 2014 and Martin et al. 2020 discuss iron and other nutrient export from the Watson River. Hawkings et al. 2020 provides data from the Leverett catchment that feeds into the main meltwater river samples. These should be included as key regional studies, and are more relevant than the comparison to the catchment in Svalbard.
We added the following:

(Line 260) **Additional support for Leverett Glacier as a key source comes from Yde et al. (2014), who estimated annual Fe export from the Watson River between 15,000 and 52,000 tons. Martin et al. (2020) further showed that glacial streams in the region, including those feeding into Watson River, deliver significantly higher DIN and PO4 than deglaciated streams, with iron concentrations that were comparable and substantial. These findings reinforce the idea that the elevated dFe levels observed here are largely driven by upstream inputs rather than solely by in-stream processes, which are unlikely to explain a near tenfold increase in concentration along the river.**

Line 276: Low compared to oceanic water? This is not surprising given ocean water concentrations for these elements are commonly much higher than freshwater endmembers (because of their long residence times). How do these compare to other rivers or other glacier meltwater studies? That would be a much more helpful comparison.

We included a comparison with other regional studies in Greenland glaciers and fjords (see below):

(Line 278) **The low concentration farther out in the fjord was in general accordance with observations in fjords and coastal waters along west Greenland. Hopwood et al. (2016) observed relatively high concentrations (13 µg L−1) in low-saline water near a glacier and river outlets in Godthåbsfjord (∼ 65◦N) and lower values of ∼2 µg L−1 near280 the fjord mouth. van Genuchten et al. (2021) observed similarly high dFe-values (13 µg L−1) within 10 km of a river outlet in Ameralik fjord (a neighboring fjord to Godthåbsfjord) while their observations around Disko Island (∼ 69◦N), i.e., an area close to coastal water masses, showed significantly lower values of ∼0.3 µg L−1.**

However, we also added the following:

(Line 291) **Compared to concentrations on the adjacent shelf, fjord values were elevated. Campbell and Yeats (1982) reported surface values (10 m depth) of 4.4, 0.7 and 1.0 µg L−1 for dMn, dNi and dCu, respectively, on the shelf off Kangerlussuaq fjord (67.75 ◦N, 57.08◦W). This study found even higher levels of dMn, dNi and dCu in the fjord (Table 1).**

Lines 288-290: This is really interesting and this discussion should be developed more - more should be made out of this finding as the idea has proved controversial in the literature. Hawkings et al. 2017 below investigated Si in this region and attributed surface fjord increases to the weathering of amorphous Si. You also cite this paper in your introduction, but it should be discussed here in the context of increasing dissolved Si at low salinities and dissolution of rock flour upon mixing with seawater.

While the distribution of silicate provides valuable insights, as discussed in Hawkings et al. 2017, our current study focuses on trace metals. We plan to investigate the role of silicate in a future study. Although weathering may contribute to the observed silicate patterns, our

current dataset is insufficient to confirm this mechanism. We added the following to acknowledge the importance of silicate weathering in this region:

Line 311: **Silicate concentrations, by contrast, were significantly higher in the fjord than in the river, pointing to internal sources. These may be tied to the weathering of glacially derived fine material (GRF), as suggested by Hawkings et al. (2017), or biological cycling. A silicate minimum between the surface and 30 m supports the idea of uptake by diatoms. Silicate weathering may play an important role in shaping nutrient and trace metal dynamics in this region, highlighting the need for further investigation into its contribution.**

Figure 10: Are there some measurements missing here? It looks like there were no samples taken between ~10 and 40 km from the ice margin?

That is correct, no SSC samples were taken at those points. We added an explanation in the legend as follows:
**Note that no SSC samples were taken between ~10 and 40 km from the ice margin.**

**Table A2.** Trace metal data ($\mu$g L$^{-1}$); iron (Fe), manganese (Mn), cobalt (Co), copper (Cu), zinc (Zn), nickel (Ni), molybdenum (Mo), arsenic (As), vanadium (V), and uranium (U). Values below quantification limit are shown as: '< quantification limit value', for each element respectively. Chromium (Cr) and lead (Pb) presented mostly all values below quantification limit except for station 17 and 35, where Cr was 0.52 and 3.25 $\mu$g L$^{-1}$, respectively. Pb was 0.10 $\mu$g L$^{-1}$ at station 17. (*) Two samples were taken at stations 6, 7, 19 and 20. Instrument uncertainties are shown in Table S1.

| Station | Lat (°N) | Lon (°W) | Fe | Mn | Co | Cu | Zn | Ni | Mo | As | V | U |
|---|---|---|---|---|---|---|---|---|---|---|---|---|
| 6 | 66.879 | -51.220 | 5.11 | 7.75 | <0.10 | 1.27 | 2.20 | 0.84 | 1.00 | <0.50 | <0.50 | 0.12 |
| 6* | 66.879 | -51.220 | <5.00 | 2.35 | <0.10 | 0.73 | 1.16 | 0.66 | 4.67 | 0.68 | <0.50 | 1.01 |
| 7 | 66.862 | -51.280 | <5.00 | 3.89 | <0.10 | 1.04 | 9.66 | 0.73 | 3.26 | 0.53 | <0.50 | 0.68 |
| 7* | 66.862 | -51.280 | <5.00 | 3.62 | <0.10 | 0.82 | 8.92 | 0.82 | 3.34 | 0.51 | <0.50 | 0.62 |
| 8 | 66.938 | -50.976 | 17.54 | 5.97 | 0.11 | 1.87 | 93.97 | 0.76 | 1.51 | <0.50 | 0.51 | 0.25 |
| 9 | 66.945 | -50.958 | <50 | 4.96 | <0.10 | 1.02 | 18.58 | 0.69 | 2.01 | <0.50 | <0.50 | 0.46 |
| 10 | 66.947 | -50.939 | <5.00 | 4.28 | <0.1 | 1.12 | 14.78 | 0.80 | 2.91 | <0.50 | <0.50 | 0.64 |
| 11 | 66.944 | -50.920 | <5.00 | 3.89 | <0.10 | 1.11 | 7.29 | 0.71 | 3.26 | <0.50 | <0.50 | 0.53 |
| 13 | 66.947 | -50.883 | 16.08 | 8.06 | 0.12 | 1.21 | 52.95 | 0.62 | 0.57 | <0.50 | 0.65 | <0.10 |
| 14 | 66.946 | -50.902 | 40.93 | 9.87 | 0.16 | 2.49 | 35.17 | 0.78 | 0.37 | <0.50 | 0.60 | <0.10 |
| 15 | 66.945 | -50.921 | 75.2 | 12.47 | 0.16 | 3.12 | 18.68 | 0.84 | 0.22 | <0.50 | 0.68 | <0.10 |
| 16 | 66.944 | -50.943 | 94.77 | 12.88 | 0.22 | 4.12 | 31.25 | 0.78 | 0.22 | <0.50 | 0.69 | <0.10 |
| 17 | 66.943 | -50.960 | 75.24 | 10.26 | 0.18 | 3.45 | 60.31 | 0.93 | 0.34 | <0.50 | 0.76 | <0.10 |
| 18 | 66.938 | -50.980 | 13.67 | 6.63 | <0.10 | 1.20 | 121.78 | 0.64 | 0.65 | <0.50 | 0.79 | <0.10 |
| 19* | 66.931 | -50.996 | <5.00 | 4.59 | <0.10 | 0.90 | 12.46 | 0.87 | 1.76 | <0.50 | <0.50 | 0.38 |
| 19 | 66.931 | -50.996 | <5.00 | 3.67 | <0.10 | 0.76 | 10.65 | 0.95 | 2.20 | <0.50 | <0.50 | 0.47 |
| 20 | 66.925 | -51.010 | <5.00 | 4.48 | <0.10 | 0.97 | 49.93 | 0.88 | 2.27 | <0.50 | <0.50 | 0.47 |
| 20* | 66.925 | -51.010 | <5.00 | 3.64 | <0.10 | 1.01 | 18.66 | 0.95 | 2.48 | <0.50 | <0.50 | 0.52 |
| 21 | 66.912 | -51.047 | <5.00 | 7.63 | <0.10 | 0.76 | 5.43 | 0.61 | 0.70 | <0.50 | 0.59 | <0.10 |
| 22 | 66.948 | -50.903 | 8.97 | 8.43 | <0.10 | 0.95 | 26.31 | 0.79 | 0.37 | <0.50 | 0.58 | <0.10 |
| 23 | 66.943 | -50.946 | <5.00 | 7.72 | 0.11 | 0.77 | 3.73 | 0.73 | 0.65 | <0.50 | 0.64 | <0.10 |
| 24 | 66.936 | -50.984 | <5.00 | 7.84 | <0.10 | 0.68 | 2.47 | 0.62 | 0.58 | <0.50 | 0.81 | <0.10 |
| 25 | 66.926 | -51.016 | <5.00 | 7.60 | <0.10 | 0.72 | 8.90 | 0.62 | 0.55 | <0.50 | 0.66 | <0.10 |
| 26 | 66.915 | -51.051 | <5.00 | 6.80 | <0.10 | 0.78 | 21.95 | 0.71 | 0.81 | <0.50 | 0.67 | 0.14 |
| 27 | 66.904 | -51.088 | <5.00 | 5.53 | <0.10 | 0.77 | 5.65 | 0.90 | 1.33 | <0.50 | <0.50 | 0.24 |
| 28 | 67.105 | -50.216 | 16.02 | 9.20 | <0.10 | 1.29 | 27.9 | 0.65 | <0.10 | <0.50 | <0.50 | <0.10 |
| 30 | 67.151 | -50.040 | 12.50 | 2.92 | <0.10 | 0.58 | 3.96 | 0.40 | <0.10 | <0.50 | <0.50 | <0.10 |
| 31 | 67.147 | -50.108 | 22.20 | 10.21 | 0.12 | 1.09 | 37.38 | 0.60 | <0.10 | <0.50 | 0.51 | <0.10 |
| 32 | 67.143 | -50.124 | 25.04 | 3.05 | <0.10 | 2.56 | 34.93 | 1.54 | <0.10 | <0.50 | <0.50 | <0.10 |
| 33 | 67.006 | -50.678 | 37.16 | 9.33 | <0.10 | 1.58 | 27.43 | 0.46 | 0.27 | <0.50 | 1.00 | <0.10 |
| 34 | 67.063 | -50.376 | 81.32 | 9.04 | 0.14 | 2.23 | 48.15 | 0.69 | 0.19 | <0.50 | 0.77 | <0.10 |
| 35 | 67.028 | -50.584 | 128.94 | 10.26 | 0.18 | 3.1 | 18.52 | 0.87 | 0.20 | <0.50 | 0.78 | <0.10 |

Table S1. Instrument uncertainties, measured as the Reference Standard, RSD, %) from all trace metal samples ($\mu g\ L^{-1}$); iron (Fe), manganese (Mn), cobalt (Co), copper (Cu), zinc (Zn), nickel (Ni), molybdenum (Mo), arsenic (As), vanadium (V), and uranium (U). RSD is not reported for values below quantification limit. Quantification limit for each element is given in the second row for each element.

| Station | 56 Fe [He] 5 µg/l | % | 55 Mn [He] 0.5 µg/l | % | 59 Co [He] 0.1 µg/l | % | 66 Zn [He] 1 µg/l | % | 60 Ni [He] 0.1 µg/l | % | 95 Mo [He] 0.1 µg/l | % | 75 As [He] 0.5 µg/l | % | 51 V [He] 0.5 µg/l | % | 238 U [No Gas] 0.1 µg/l | % |
|---|---|---|---|---|---|---|---|---|---|---|---|---|---|---|---|---|---|---|
| | [ug/l] | RSD | [ug/l] | RSD | [ug/l] | RSD | [ug/l] | RSD | [ug/l] | RSD | [ug/l] | RSD | [ug/l] | RSD | [ug/l] | RSD | [ug/l] | RSD |
| 6 | 5.11 | 8.93 | 7.75 | 6.26 | 0.10 | 5.21 | 2.20 | 17.05 | 0.84 | 5.56 | 1.00 | 4.00 | <QL | | <QL | | 0.12 | 3.98 |
| 6 | <QL | | 2.36 | 4.41 | <QL | | 1.16 | 11.88 | 0.66 | 8.72 | 4.67 | 6.87 | 0.68 | 14.25 | <QL | | 1.01 | 1.79 |
| 7 | <QL | | 3.89 | 6.95 | <QL | | 9.66 | 6.97 | 0.73 | 14.35 | 3.26 | 2.46 | 0.53 | 27.35 | <QL | | 0.68 | 1.77 |
| 7 | <QL | | 3.62 | 2.88 | <QL | | 8.92 | 8.52 | 0.82 | 3.49 | 3.34 | 5.45 | 0.51 | 16.96 | <QL | | 0.62 | 1.55 |
| 8 | 17.54 | 1.91 | 5.97 | 1.42 | 0.11 | 4.88 | 93.97 | 1.61 | 0.76 | 14.56 | 1.51 | 6.96 | <QL | | 0.51 | 7.52 | 0.25 | 2.70 |
| 9 | <QL | | 4.96 | 1.65 | <QL | | 18.58 | 4.60 | 0.69 | 18.39 | 2.01 | 6.11 | <QL | | <QL | | 0.46 | 1.96 |
| 10 | <QL | | 4.28 | 2.74 | <QL | | 14.78 | 3.76 | 0.80 | 14.94 | 2.91 | 4.86 | <QL | | <QL | | 0.64 | 1.54 |
| 11 | <QL | | 3.89 | 16.13 | <QL | | 7.29 | 9.59 | 0.71 | 6.39 | 3.26 | 7.06 | <QL | | <QL | | 0.53 | 3.48 |
| 13 | 16.08 | 3.39 | 8.06 | 1.56 | 0.12 | 7.19 | 52.95 | 5.39 | 0.62 | 3.34 | 0.57 | 4.88 | <QL | | 0.65 | 28.62 | <QL | |
| 14 | 40.93 | 9.66 | 9.87 | 10.18 | 0.16 | 13.51 | 35.17 | 13.65 | 0.78 | 14.48 | 0.37 | 5.11 | <QL | | 0.60 | 6.48 | <QL | |
| 15 | 75.20 | 3.47 | 12.47 | 5.59 | 0.16 | 9.69 | 18.68 | 4.34 | 0.84 | 10.20 | 0.22 | 19.68 | <QL | | 0.68 | 1.52 | <QL | |
| 16 | 94.77 | 7.76 | 12.88 | 9.04 | 0.22 | 6.40 | 31.25 | 11.18 | 0.78 | 6.84 | 0.22 | 15.25 | <QL | | 0.69 | 2.76 | <QL | |
| 17 | 75.24 | 4.85 | 10.26 | 8.31 | 0.18 | 9.21 | 60.31 | 3.01 | 0.93 | 4.84 | 0.34 | 12.88 | <QL | | 0.76 | 17.60 | <QL | |
| 18 | 13.67 | 4.65 | 6.63 | 4.88 | <QL | | 121.78 | 7.04 | 0.64 | 14.32 | 0.65 | 14.33 | <QL | | 0.79 | 8.31 | <QL | |
| 19 | <QL | | 4.59 | 12.94 | <QL | | 12.46 | 4.02 | 0.87 | 4.20 | 1.76 | 4.18 | <QL | | <QL | | 0.38 | 5.41 |
| 19 | <QL | | 3.67 | 2.59 | <QL | | 10.65 | 9.96 | 0.95 | 9.60 | 2.20 | 0.61 | <QL | | <QL | | 0.47 | 3.03 |
| 20 | <QL | | 4.48 | 10.45 | <QL | | 49.93 | 7.82 | 0.88 | 10.75 | 2.27 | 1.87 | <QL | | <QL | | 0.47 | 1.39 |
| 20 | <QL | | 3.64 | 2.82 | <QL | | 18.66 | 0.70 | 0.95 | 7.13 | 2.48 | 4.83 | <QL | | <QL | | 0.52 | 2.37 |
| 21 | <QL | | 7.63 | 9.11 | <QL | | 5.43 | 8.41 | 0.61 | 7.53 | 0.70 | 8.28 | <QL | | 0.59 | 4.51 | <QL | |
| 22 | 8.97 | 7.09 | 8.43 | 10.73 | 0.10 | 10.99 | 26.31 | 12.42 | 0.79 | 6.18 | 0.37 | 10.08 | <QL | | 0.58 | 9.67 | <QL | |
| 23 | <QL | | 7.72 | 2.54 | 0.11 | 8.02 | 3.73 | 9.54 | 0.73 | 12.07 | 0.65 | 16.09 | <QL | | 0.64 | 32.60 | <QL | |
| 24 | <QL | | 7.84 | 0.06 | <QL | | 2.47 | 23.82 | 0.62 | 12.83 | 0.58 | 2.89 | <QL | | 0.81 | 23.29 | <QL | |
| 25 | <QL | | 7.60 | 6.21 | <QL | | 8.90 | 6.29 | 0.62 | 12.84 | 0.55 | 9.85 | <QL | | 0.66 | 15.47 | <QL | |
| 26 | <QL | | 6.80 | 5.66 | <QL | | 21.95 | 3.17 | 0.71 | 16.64 | 0.81 | 2.39 | <QL | | 0.67 | 5.97 | 0.14 | 1.52 |
| 27 | <QL | | 5.53 | 0.99 | <QL | | 5.65 | 7.77 | 0.90 | 0.72 | 1.33 | 2.64 | <QL | | <QL | | 0.24 | 3.39 |
| 28 | 16.02 | 10.34 | 9.20 | 8.64 | <QL | | 27.90 | 7.17 | 0.65 | 6.01 | <QL | | <QL | | <QL | | <QL | |
| 30 | 12.50 | 4.92 | 2.92 | 2.34 | <QL | | 3.96 | 7.50 | 0.40 | 1.87 | <QL | | <QL | | <QL | | <QL | |
| 31 | 22.20 | 5.44 | 10.21 | 3.76 | 0.12 | 13.49 | 37.38 | 4.41 | 0.60 | 9.64 | <QL | | <QL | | 0.51 | 9.96 | <QL | |
| 32 | 25.04 | 19.91 | 3.05 | 23.24 | <QL | | 34.93 | 18.76 | 1.54 | 22.12 | <QL | | <QL | | <QL | | <QL | |
| 33 | 37.16 | 4.53 | 9.33 | 2.52 | <QL | | 27.43 | 5.62 | 0.46 | 13.68 | 0.27 | 20.85 | <QL | | 1.00 | 2.97 | <QL | |
| 34 | 81.32 | 3.52 | 9.04 | 1.74 | 0.14 | 15.09 | 48.15 | 3.57 | 0.69 | 17.39 | 0.19 | 22.79 | <QL | | 0.77 | 0.80 | <QL | |
| 35 | 128.94 | 9.22 | 10.26 | 11.97 | 0.18 | 4.11 | 18.52 | 12.82 | 0.87 | 7.53 | 0.20 | 12.92 | <QL | | 0.78 | 6.41 | <QL | |